# Heterochromatin epimutations impose mitochondrial dysfunction to confer antifungal resistance

Andreas Fellas [ID][1,2,4], Alison L Pidoux [ID][1,4✉], Pin Tong[1], Harriet H Hewes [ID][1], Emma C Wallace [ID][1,3] & Robin C Allshire [ID][1✉]

## Abstract

**Antifungal resistance in pathogenic fungi endanger global health and food supply. Wild-type fission yeast, _Schizosaccharomyces pombe_, can gain resistance to insults including caffeine and antifungal compounds through reversible epimutations. Resistant epimutants exhibit ectopic histone-H3K9 methylation-dependent heterochromatin islands, repressing underlying genes. Two genes whose heterochromatin island-induced repression causes resistance encode mitochondrial proteins: LYR-domain protein Cup1 and Cox1 translation regulator Ppr4. Genetic mutations, _cup1-tt_ and _ppr4Δ_, that phenocopy epimutants, cause mitochondrial dysfunction, including respiratory deficiency, poor growth on non-glucose carbon sources, and elevated reactive oxygen species. Transcriptomic analyses indicate _cup1-tt_ and _ppr4Δ_ cells activate Pap1 transcription factor-dependent oxidative stress response and mitonuclear retrograde pathways. Pap1 nuclear localisation and recruitment to promoters of oxidoreductase and membrane transporter genes is increased, causing increased efflux activity. _cup1_ and _ppr4_ epimutants likewise show mitochondrial dysfunction phenotypes and increased efflux, explaining how heterochromatin-island epimutations cause drug resistance. Thus, wild-type cells harness epimutations that impose mitochondrial dysfunction to bypass external insults. As mitochondrial dysfunction is linked to antifungal resistance in several fungi, similar epimutations likely contribute to development of resistance in fungal pathogens.**

**Keywords** Epigenetics; Bet-hedging; Electron Transport; ROS; Oxidative Stress
**Subject Categories** Chromatin, Transcription & Genomics; Microbiology, Virology & Host Pathogen Interaction; Organelles

## Introduction

Microorganisms including fungi experience ever-changing environments. Adverse environmental conditions include both abiotic and biotic stresses such as antifungal drugs. Cell survival depends on their ability to adapt to new or fluctuating environments by changing phenotype. Strategies to deal with environmental challenges vary in rapidity of implementation and longevity of impact (Hernandez-Elvira and Sunnerhagen, 2022; Sabarís et al, 2023). Environmental stress can induce immediate responses through transcriptional changes, enabling cells to adapt to and survive external insults (Chen et al, 2008, 2003; López-Maury et al, 2008; Papadakis and Workman, 2015; Yaakoub et al, 2022). Such changes in transcriptional profile are generally not heritable and disappear once the insult is removed. In contrast, DNA-based changes such as point mutations are essentially permanent and provide adaptation in a miniscule proportion of a starting population, generating mutant cells that tolerate the new environment. Adaptation to challenging environments may come with fitness costs and alleles that are beneficial or neutral in one environment may be harmful or deleterious in the original or alternative environments. Genetic mutations are poorly suited to fast adaptation as they arise and revert infrequently. The essentially irreversible nature of most genetic mutations offers stable but inflexible phenotypic adaptation.

Semi-stable changes that provide adaptation to the new environment but can be reversed at relatively high frequency (orders of magnitude higher than reversion of a point mutations) upon return to the original environmental conditions offer 'best-of-both worlds' flexibility in coping with changing environments (Sabarís et al, 2023). Some semi-stable changes are DNA-based, such as aneuploidy (Berman, 2016). The relatively high rate at which aneuploidy can arise (compared to point mutations) combined with facile loss of extra chromosomes, makes aneuploidy a convenient adaptive mechanism in fluctuating environments (Vande Zande et al, 2023).

Epigenetic mechanisms can also give rise to semi-stable phenotypes which are advantageous in stressful environments but can dissipate if conditions change. The term 'epigenetic' is used here to denote mechanisms in which distinct activity states operate without alteration of DNA sequence and which are heritable through cell division independently of any original initiating signal (Cavalli and Heard, 2019). Epigenetic inheritance mechanisms also include prions, protein-based epigenetic agents. Abundant evidence in _Saccharomyces cerevisiae_ indicates prions can profoundly affect phenotype (Byers and Jarosz, 2014; Oamen et al, 2020). However, in most microorganisms,

---

[1]Centre for Cell Biology, Institute of Cell Biology, School of Biological Sciences, The University of Edinburgh, Edinburgh EH9 3BF Scotland, UK. [2]Present address: MRC Human Genetics Unit, Institute of Genetics and Cancer, The University of Edinburgh, Edinburgh, UK. [3]Present address: Cancer Research UK Cambridge Institute, Li Ka Shing Centre, Cambridge, UK. [4]These authors contributed equally: Andreas Fellas, Alison L Pidoux. ✉E-mail: alison.pidoux@ed.ac.uk; robin.allshire@ed.ac.uk

documented epigenetic mechanisms involve DNA methylation or histone post-translational modifications (PTMs) (Fitz-James and Cavalli, 2022; Moazed, 2011). In contrast to mutation rate, which is relatively low, DNA and histone modifications with epigenetic potential can occur rapidly. Chromatin-based epigenetic mechanisms differ from short-duration immediate transcriptional responses to environmental changes in that they are heritable. In theory epimutations can be easily installed but easily reversed, offering adaptive plasticity in fluctuating environments. Semi-stable adaptive states—both genetic and epigenetic—offer populations bet-hedging strategies to cope with changing environments by enhancing phenotypic diversity (Sabarís et al, 2023).

Several examples of epigenetic adaptation have been described in unicellular prokaryotes and eukaryotes. DNA methylation-based epigenetic mechanisms affect pathogenicity and antibiotic resistance in bacteria (Villalba de la Peña and Kronholm, 2024). Semi-stable RNAi-based epimutations providing resistance to the antifungal agents FK506 and rapamycin occur in the pathogenic fungus *Mucor circinelloides* (Calo et al, 2014). In the fission yeast, *Schizosaccharomyces pombe*, heterochromatin-based epimutations have been isolated that are unstably resistant to caffeine (Torres-Garcia et al, 2020b).

Heterochromatin in *S. pombe* is dependent on a single histone H3 Lysine 9 (H3K9) methyltransferase, Clr4. The main blocks of heterochromatin are the pericentromeric outer repeats, subtelomeric regions and the silent mating type locus where it performs structural and silencing roles. In addition, *S. pombe* genes embedded in H3K9 methylation-dependent (H3K9me) heterochromatin are transcriptionally repressed (Allshire and Madhani, 2018; Grewal, 2023). The installation of H3K9me by Clr4 is counteracted by anti-silencing factors: the histone acetyltransferase Mst2 and the H3K9-demethylase Epe1 (Wang et al, 2015; Yaseen et al, 2022; Zofall et al, 2016, 2012; Zofall and Grewal, 2006). Loss of these anti-silencing factors leads to extensive ectopic deposition of heterochromatin at euchromatic sites (Larkin et al, 2024; Wang et al, 2015; Zofall et al, 2012). H3K9me-dependent heterochromatin domains can be inherited through a reader-writer coupling mechanism when the function of the demethylase Epe1 is perturbed (Audergon et al, 2015; Ragunathan et al, 2015).

In wild-type *S. pombe* cells, unstable caffeine-resistant isolates arise which contain H3K9me2-heterochromatin islands at novel genomic locations (Torres-Garcia et al, 2020b). In such epimutants, resistance results from the decreased expression of genes located within heterochromatin islands, without alteration of the underlying DNA sequence. Unlike genetic mutants which retain resistance when grown non-selectively, epimutants are semi-stable, retaining resistance over many generations, but gradually losing resistance upon extended non-selective growth in the absence of the external insult. Heterochromatin-mediated resistance has been recapitulated by tethering TetR-Clr4 via tetO operators to create synthetic heterochromatin at naïve epimutation loci (Torres-Garcia et al, 2020b). Thus, epimutation formation provides a bet-hedging mechanism within *S. pombe* wild-type populations so that under adverse conditions, a proportion of cells exhibit distinct characteristics which allow them to survive (Torres-Garcia et al, 2020b).

Previously, the heterochromatin island in the UR1 caffeine-resistant epimutant was shown to reduce expression of the *hba1+* gene (Torres-Garcia et al, 2020b) which is required for the continuous nuclear export of the Pap1 transcription factor, a key mediator of the oxidative stress response. Cells lacking Hba1 exhibit multidrug resistance due to Pap1-induced increased expression of genes encoding antioxidants and cellular efflux processes (Castillo et al, 2003). Thus, heterochromatin-mediated repression of *hba1+* likely mediates resistance through upregulation of efflux pumps via the Pap1 stress-response pathway. In numerous epimutants (UR2, UR8 to UR30), a heterochromatin island impinges on the promoter region of the essential *cup1+* gene (Torres-Garcia et al, 2020b), which encodes a mitochondrial LYR protein predicted to be an assembly factor for mitochondrial electron transport chain (ETC) complexes (Angerer, 2015). A promoter impaired *cup1-tt* allele also mediates resistance to caffeine, consistent with *cup1+* repression being the source of resistance phenotypes in the UR2 epimutant (Torres-Garcia et al, 2020b). However, it is not understood how *cup1+* repression confers resistance to caffeine. For epimutations at other loci (UR3-UR6), the gene or genes whose repression causes caffeine resistance remain to be identified.

UR1 and UR2 epimutants also show resistance to azole-based antifungal compounds (Torres-Garcia et al, 2020b). Pathogenic fungi cause numerous human diseases (Fisher et al, 2022) and threaten food security and biodiversity (Fones et al, 2020). Resistance to antifungals is rising, involving mechanisms such as target site mutation, efflux pump upregulation, and genomic plasticity (Fisher et al, 2022, 2018). Understanding these mechanisms is critical for design and deployment of antifungal agents.

This study examines how epimutations which reduce gene expression of specific chromosomal regions in *S. pombe* confer resistance to external insults such as caffeine and a key class of antifungals that target the Cyp51/Erg11 lanosterol 14-α demethylase (generally known as Azoles or DMIs—DeMethylase Inhibitors). *ppr4+*, which also encodes a mitochondrial protein (Kühl et al, 2011), is identified as the key repressed gene within the UR3 heterochromatin island epimutation that confers resistance. Analysis of genetic mutants that phenocopy the UR2 and UR3 epimutants demonstrates that cells with reduced or absent expression of these mitochondrial proteins display defective mitochondrial function and increased reactive oxygen species (ROS). Transcriptional profiling reveals that both the mito-nuclear retrograde (MNR) and the Pap1-dependent oxidative stress response pathways are induced in these mutants, resulting in the upregulation of genes encoding antioxidants, membrane transporters, and thereby increased efflux. Cells lacking other mitochondrial proteins, such as electron transport chain components, exhibit similar phenotypes, including resistance to insults. The UR2 (*cup1+*) and UR3 (*ppr4+*) epimutations themselves share these features, indicating that heterochromatin-mediated repression of mitochondrial protein genes causes mitochondrial dysfunction, including increased ROS, leading to transcriptional changes that enable cells to exploit enhanced efflux to impart resistance to insults and antifungal compounds.

# Results

## Identification of genes whose repression in epimutants confers caffeine resistance

In unstable caffeine-resistant epimutants the ectopic heterochromatin islands decrease expression of underlying genes (Torres-

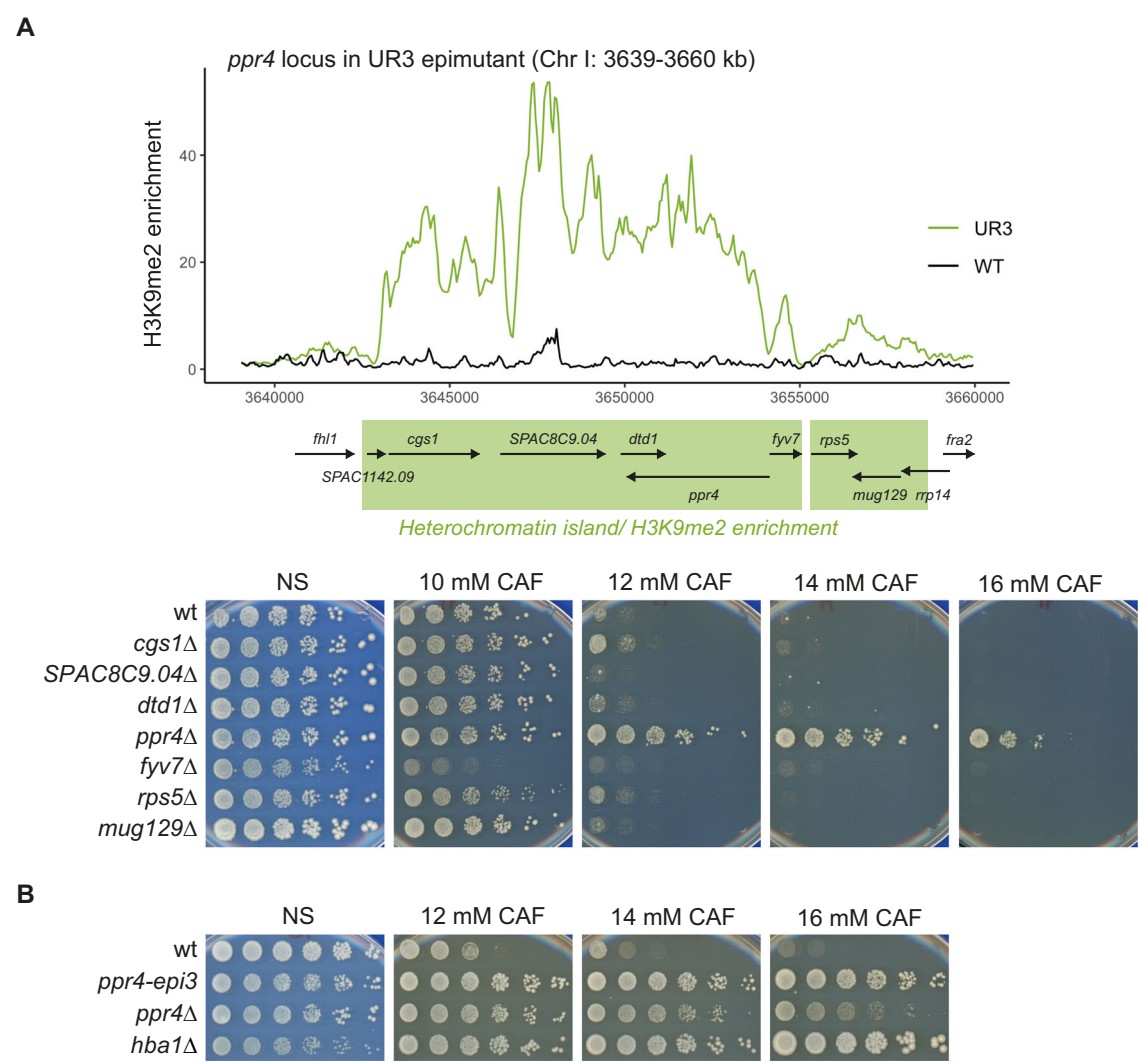

**Figure 1. Deletion of candidate genes within the UR3 island region identifies *ppr4Δ* as caffeine resistant.**

(A) Genes within UR3 heterochromatin island. Upper: Schematic of region of Chr I (3639–3660 kb) which contains the UR3 epimutation heterochromatin island. H3K9me2 ChIP-Seq data plotted from GEO: GSE138436; GSM4107922 (Torres-Garcia et al, 2020b). Protein-coding genes are indicated. Green shading indicates approximate extent of heterochromatin island in UR-3. Lower panel: Growth assays to assess resistance to caffeine of gene deletion strains. Five-fold serial dilutions of indicated strains spotted onto non-selective plates (NS; YES media) or plates containing caffeine (CAF) at the indicated concentrations. wt, wild-type. Plates photographed after 2–8 days at 32 °C. (B) Growth assay to compare resistance of *ppr4* mutant and epimutant. wt, wild-type; *hba1Δ* is a resistant control strain. Performed as in (A). Source data are available online for this figure.

Garcia et al, 2020b). Deletion of *hba1+* or downregulation of *cup1+* through promoter manipulation demonstrated that it is repression of these specific genes within UR1 and UR2, respectively, that mediates resistance (Torres-Garcia et al, 2020b). A similar approach was taken with the other caffeine-resistant epimutants to pinpoint genes whose decreased expression confers resistance. The heterochromatin island in the UR3 epimutant spans ~8 kb, encompassing genes encoding proteins Cgs1, SPAC8C9.04, Dtd1, Ppr4, Fyv7, Rps5, and Mug129 interspersed with 12 non-coding RNA genes (Fig. 1A).

To identify gene(s) responsible for the caffeine resistance in the UR3 island locus, each protein-coding gene was deleted in 972 h⁻ wild-type cells and the ability of resultant strains to grow on various insults tested in serial dilution plating assays. Deletion of

*ppr4+* resulted in resistant growth on plates containing 16 mM caffeine (CAF), a concentration at which the UR3 epimutant also grows (Fig. 1A). Deletion of other genes residing within the UR3 island resulted in minimal caffeine resistance, suggesting that *ppr4+* is the key gene in the UR3 island whose repression mediates caffeine resistance. A similar candidate gene approach was taken for the remaining epimutants, UR4, UR5 and UR6. The *fio1+* and *mbx2+* genes showed large reductions in their mRNA levels in the UR5 and UR6 epimutant isolates, respectively (Torres-Garcia et al, 2020b). However, deletion of these or other single genes within the domains covered by the UR4, UR5, or UR6 islands did not impart a caffeine-resistant phenotype (Fig. EV1A–C). As the heterochromatin islands formed in the UR4, UR5, and UR6 epimutants encompass many genes it is possible that downregulation of several

genes within a single island is required to elicit resistance. Therefore, larger regions of ~7–14 kb containing 3–6 genes were deleted within the UR4 island domain and, in some cases, this resulted in a moderate increase in caffeine and azole resistance; $hba1\Delta$ cells were highly resistant as expected (Castillo et al, 2003) (Fig. EV1B,C). These analyses indicate that the simultaneous repression of multiple genes in the UR4, UR5, and UR6 islands may provide resistance to caffeine. However, deletion of $ppr4^+$ clearly phenocopies the UR3 epimutant—henceforward $ppr4\text{-}epi3$—resistance phenotype (Fig. 1B), demonstrating that $ppr4^+$ is a gene whose repression through heterochromatin island formation mediates caffeine resistance.

## Cup1 or Ppr4 deficiency causes mitochondrial dysfunction

Intriguingly, both Cup1 and Ppr4 are nuclear-encoded mitochondrial proteins. Cup1 harbours a Leu/Tyr/Arg (LYR) domain that in other eukaryotes is associated with assembly factors and accessory subunits of the electron transport chain (ETC) (Angerer, 2015). Ppr4 is known to specifically activate the translation of mitochondrially-encoded $cox1^+$ transcripts to produce cytochrome c oxidase subunit 1, a key component of mitochondrial ETC complex IV (Herbert et al, 2021; Kühl et al, 2011). S. pombe cells primarily utilise fermentation to generate energy when grown in glucose-rich YES media (Klein et al, 2013; Klement et al, 2011). The respiratory capacity of $cup1\text{-}tt$ and $ppr4\Delta$ cells was tested by growing them on alternative carbon sources i.e. glycerol- or galactose-containing media. Media lacking glucose prevents or reduces fermentation, forcing cells to use respiration, generating metabolites and ATP via the tricarboxylic acid cycle and ETC in mitochondria (Chiron et al, 2007). Growth of fission yeast lacking components of electron transport chain (ETC) complexes was impaired ($cox4\Delta$; complex IV) or completely inhibited ($ndi1\Delta$, complex I; $qcr7\Delta$, complex III; $atp2\Delta$, complex V) on media containing glycerol or galactose as the main or sole carbon source (Fig. 2A). Likewise, $ppr4\Delta$ cells were unable to grow on glycerol or galactose, as reported previously (Su et al, 2017). The growth of $cup1\text{-}tt$ cells on glycerol or galactose media was also impaired, indicating that respiration is defective when $cup1^+$ expression is decreased by approximately half (Torres-Garcia et al, 2020b). To further evaluate respiratory competence, wild-type and mutant cells were exposed to the colourless redox indicator 2,3,5-triphenyltetrazolium chloride (TTC) which is reduced to red-coloured 2,3,5-triphenyltetrazolium formazan (TPF) by cells undergoing respiration (Nagai et al, 1961; Ogur et al, 1957; Tanaka et al, 2021). Wild-type cells efficiently converted white TTC to red TPF while $cup1\text{-}tt$, $ppr4\Delta$ and some ETC mutants remained white ($qcr7\Delta$, $atp2\Delta$) or intermediate shades of pink ($ndi1\Delta$, $cox4\Delta$; Fig. 2A). Compared to wild-type, some ETC mutants, $cup1\text{-}tt$ and $ppr4\Delta$ also produced low levels of TPF in a liquid-based TTC assay (Fig. EV2A). Neither $cup1\text{-}tt$ nor $ppr4\Delta$ cells displayed gross defects in mitochondrial morphology (Fig. EV2B). Staining of S. pombe mitochondria with Mitotracker RedCMXRos has been shown to depend on mitochondrial membrane potential, whereas Mitotracker Green staining is membrane-potential independent (Uehara et al, 2021). Both $cup1\text{-}tt$ and $ppr4\Delta$, along with $qcr7\Delta$ cells appeared to have reduced membrane potential compared to wild-type as evidenced by reduced ratio red/green Mitotracker ratio by flow cytometry

(Fig. EV2C), again suggestive of defects in mitochondrial function. ETC mutants, especially $qcr7\Delta$ (complex III), displayed resistance to both caffeine and azole-based antifungals (Fig. 2B). $ppr4\Delta$ cells, along with $hba1\Delta$ and $cup1\text{-}tt$ cells, exhibited cross-resistance to the clinical azole-based antifungals Fluconazole (FLC) and Clotrimazole (CLT), and the agritech azole antifungal Prothioconazoledesthio (PRD) (Fig. 2B). Interestingly, like ETC mutants and $cup1\text{-}tt$ and $ppr4\Delta$, impairing mitochondrial function with the Complex III inhibitor Antimycin A which binds Qi site of cytochrome C and increases production of ROS (Quinlan et al, 2011) (Zuin et al, 2008), promotes resistance of wild-type cells to both caffeine and fluconazole, enabling them to grow as well as some mitochondrial mutants in the presence of these insults (Fig. EV2D).

Disruption of oxidative phosphorylation in S. pombe results in mitochondrial ETC electron leakage and generation of reactive oxygen species (ROS) (Zuin et al, 2008). Intracellular ROS can be detected with the DCFH-DA reporter that is converted to fluorescent 2′,7′-dichlorofluorescein (DCF) upon exposure to ROS (Jiang et al, 2021). Using this assay, $cup1\text{-}tt$ and $ppr4\Delta$ cells, along with cells lacking ETC complex I, III, IV or V components, but not complex II, displayed elevated ROS levels (Su et al, 2017; Zuin et al, 2008) relative to wild-type cells when assayed by fluorescence microscopy (Fig. 2C) or flow cytometry (Figs. 2D and EV2E). Thus, caffeine and azole-resistant mutants $cup1\text{-}tt$ and $ppr4\Delta$ exhibit mitochondrial dysfunction as indicated by elevated intracellular ROS levels associated with defective respiration. Reciprocally, some respiration-defective ETC mutants are also resistant to caffeine and azole antifungals. Interestingly, overexpression of the peroxidase Gpx1, a hydrogen peroxide scavenger protein (Kim et al, 2010), modestly reduced the caffeine resistance phenotype of $cup1\text{-}tt$ and $ppr4\Delta$ (Fig. EV2F), supporting a link between elevated ROS and resistance to insults.

## Cup1 and Ppr4 deficiencies result in mito-nuclear retrograde gene repression

To explore the mechanisms by which cells expressing reduced Cup1 levels or lacking Ppr4 mediate resistance to caffeine or azole-based antifungals, RNA-Seq of polyA RNA from wild-type, $cup1\text{-}tt$ and $ppr4\Delta$ cells was performed. Heat-maps of transcript levels shows that $cup1\text{-}tt$ and $ppr4\Delta$ cells are more similar to each other than either is to wild-type cells (Fig. 3A). Comparison of $cup1\text{-}tt$ and $ppr4\Delta$ transcriptomes revealed that they share 514 differentially expressed genes, 389 of which are upregulated and 125 downregulated (Fig. 3B). Functional profiling of genes affected in both $cup1\text{-}tt$ and $ppr4\Delta$ cells through GO and KEGG term analyses (Raudvere et al, 2019) are consistent with both mutants having a deleterious impact on cellular function as indicated by the induction of genes encoding oxidoreductases, iron import and cellular detoxification and the repression of genes involved in mitochondrial aerobic respiration (Fig. 3C; Table EV1).

Communication between mitochondria and nuclei is essential for coordinating the expression of different mitochondrial components encoded by either the nuclear or mitochondrial genomes. This is especially important for various proteins involved in oxidative phosphorylation of which some are encoded by each genome (Couvillion et al, 2016; Herbert et al, 2021). Previously, comparison of gene expression changes that occur upon inhibition of ETC Complex III (Cytochrome C) with antimycin A and the two

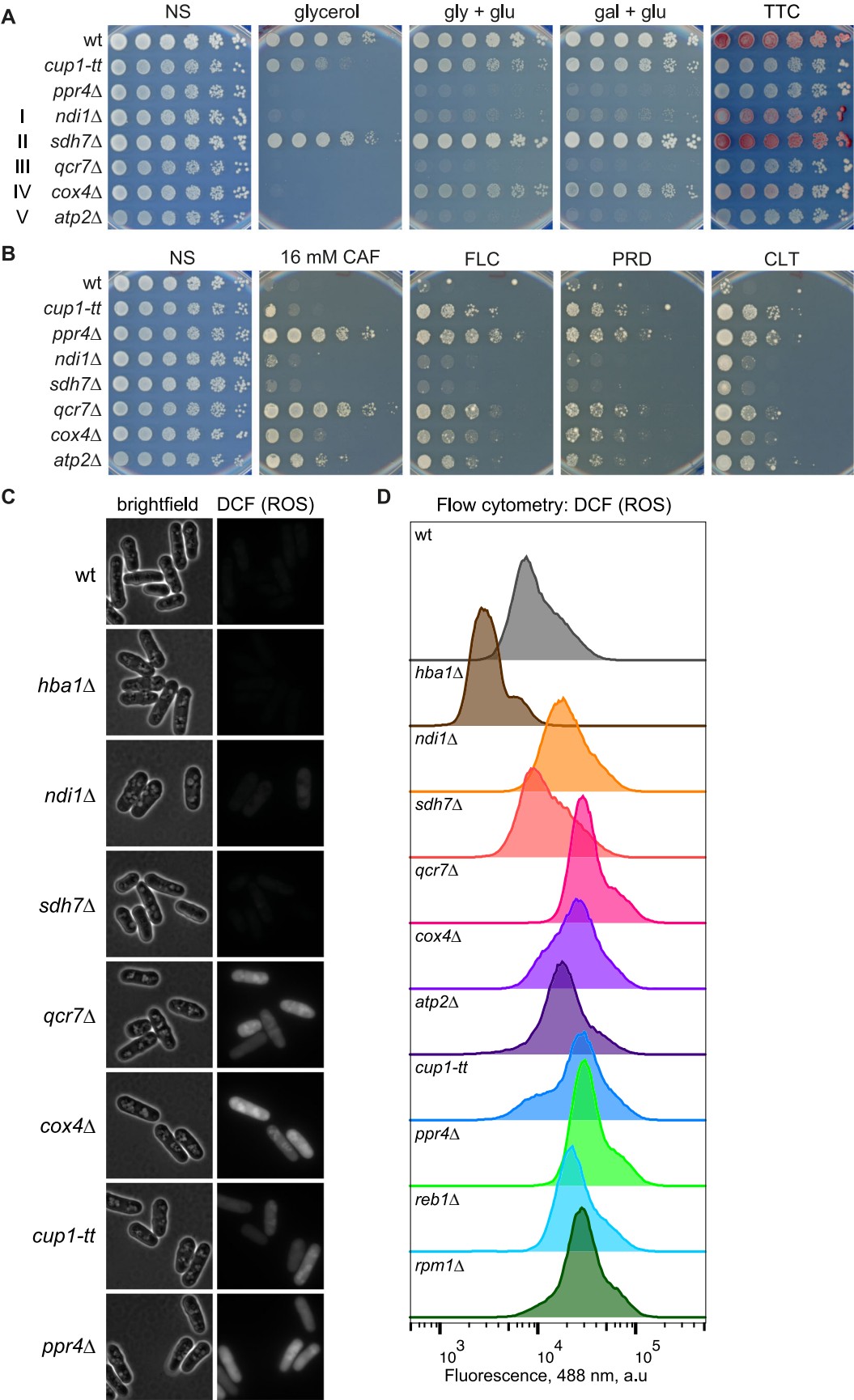

**Figure 2. Deficiency of Cup1, Ppr4 or ETC components cause mitochondrial dysfunction and resistance to insults.**

(A) Growth assay to assess mitochondrial competence. Five-fold serial dilutions of the indicated strains spotted on regular YES media containing 2% glucose, or YE plates in which glucose was replaced with 3% glycerol or 3% glycerol + 0.1% glucose (gly + glu) or 2% galactose + 0.1% glucose (gal + glu). After 3 days' growth, colonies on YES were overlaid with TTC-containing agarose and incubated for ~24 h. ETC complexes are indicated in Roman numerals at left. Red colour indicates respiratory competence. (B) Growth assay to assess growth on caffeine and antifungal drugs. Five-fold serial dilutions of indicated strains spotted onto non-selective plates (NS; YES media) or plates containing 16 mM caffeine (CAF), or azole-based antifungals: 0.3 mM fluconazole (FLC), 0.4 μM prothioconazole-desthio (PRD), 50 ng/ml clotrimazole (CLT). Plates photographed after 2–8 days at 32 °C. (C) DCHF-DA staining to assess levels of reactive oxygen species. Cells of the indicated strains were incubated in the ROS indicator DCFH-DA which is converted to fluorescent DCF in the presence of ROS, and imaged under brightfield and 488 nm illumination. Scale bar, 10 μm. (D) Assessment of ROS levels by flow cytometry. Flow cytometry profiles at 488 nm of cells of the indicated strains incubated in DCFH-DA. Source data are available online for this figure.

respiratory deficient mutants, *reb1Δ* and *rpm1Δ* defined the fission yeast mito-nuclear retrograde (MNR) response (Malecki et al, 2016). The repression of genes involved in aerobic respiration, with the exception of ETC Complex II, is a hallmark of mitochondrial dysfunction and the MNR response. RNA-Seq analysis indicates that, apart from ETC complex II, ETC components are repressed in both *cup1-tt* and *ppr4Δ* cells (Fig. 3C). These changes in gene expression in *cup1-tt* and *ppr4Δ* cells significantly overlap with the altered transcriptional profile of antimycin A-treated cells and the characterized mito-nuclear retrograde response (Figs. 4A and E-V3A,B). Changes in transcript levels detected by analysis of RNA-Seq data were confirmed by RT-qPCR analyses of nuclear-encoded ETC genes (Fig. EV3C). In common with *cup1-tt*, *ppr4Δ* and several ETC mutants, the MNR pathway-activating mutants *reb1Δ* and *rpm1Δ* cells exhibit caffeine and fluconazole resistance, impaired growth on non-fermentable carbon sources, appear white or pink in the TTC-overlay assay and have increased ROS levels, suggestive of respiratory deficiency (Figs. 2D and 4B). The overlapping changes in the transcriptional profiles and phenotypes of *cup1-tt*, *ppr4Δ*, *reb1Δ* and *rpm1Δ* cells suggest that compromised Cup1 or Ppr4 function activates the MNR response and that mitochondrial dysfunction, such as ETC disruption, promotes resistance to external insults.

## Oxidative stress response genes are induced in *cup1-tt* and *ppr4Δ* cells

In addition to the repression of genes indicative of the MNR response being activated in *cup1-tt* and *ppr4Δ* cells, it was noticeable that the genes encoding components of the core environmental stress response (CESR) (Chen et al, 2003) and core oxidative stress genes (COSG) (Chen et al, 2008) were induced in both mutants (Fig. 4C). A key player in the response of fission yeast to mild oxidative stress is the AP-1-like transcription factor, Pap1. Under non-stress conditions Pap1 localises to the cytoplasm by continual export from the nucleus, a process mediated by the karyopherin Crm1 and the RanGAP Hba1 (Toone et al, 1998). Disruption of this pathway has been implicated in multidrug resistance (Benko et al, 1998; Kumada et al, 1996; Toda et al, 1992). Mild oxidative stress results in Pap1 oxidation and its retention in the nucleus, where it upregulates expression of genes that contribute to survival in oxidative stress such as those encoding antioxidants and efflux pumps (Calvo et al, 2012; Chen et al, 2008; Papadakis and Workman, 2015). Several genes under the transcriptional control of Pap1 are upregulated in *cup1-tt* and *ppr4Δ* cells, including those encoding the transmembrane transporter Caf5 and the oxidoreductase Obr1 (Fig. 3C). To detect

upregulation of Pap1-responsive genes, GFP under the control of the Pap1-dependent *obr1⁺* promoter was expressed from a locus on chromosome 2. As expected, cells lacking the Ran-GAP Hba1 exhibited high *obr1* promoter-driven GFP expression (Fig. EV4A), due to the lack of Pap1 export from nuclei (Calvo et al, 2012). Moreover, compared to wild-type cells, increased *obr1^Pro^ > GFP* reporter expression was evident in *cup1-tt* and *ppr4Δ* cells and cells lacking Qcr7 from ETC Complex III (*qcr7Δ*; Fig. 5A). Increased *caf5* promoter-driven GFP expression (*caf5^Pro^ > GFP*) was also observed in *cup1-tt*, *ppr4Δ*, *qcr7Δ* and *hba1Δ* mutants (Fig. EV4B). Thus, the *obr1* and *caf5* Pap1-dependent promoters are activated in *cup1-tt* and *ppr4Δ* cells and cells with defective mitochondrial ETC function such as *qcr7Δ*.

To further test if *cup1-tt* and *ppr4Δ*, and other mitochondrial mutants activate the Pap1-dependent oxidative stress response pathway, cells expressing a GFP-Pap1 fusion protein from the endogenous *pap1* locus was utilised. As expected (Castillo et al, 2003), GFP-Pap1 was predominantly cytoplasmic in wild-type but constitutively localised to the nucleus in *hba1Δ* cells (Fig. 5B). *cup1-tt* and *ppr4Δ* cells also displayed a measurable increase in nuclear GFP-Pap1 levels compared to wild-type cells. Chromatin immunoprecipitation (ChIP) of GFP-Pap1 detected high levels of enrichment of GFP-Pap1 on the promoters of Pap1-responding genes in *hba1Δ* cells or cells treated with hydrogen peroxide (Fig. EV4C), as expected (Calvo et al, 2012). A more modest increase in GFP-Pap1 was detected on the *obr1⁺* promoter in *cup1-tt*, *ppr4Δ*, *qcr7Δ* and *cox4Δ* cells compared to wild-type, and to varying levels on other Pap1-dependent promoters (Fig. 5C). This increase in GFP-Pap1 at Pap1-dependent promoters is consistent with the modest activation of Pap1-dependent genes such as *obr1⁺*, *caf5⁺* and *srx1⁺* in response to defective mitochondrial function in *cup1-tt*, *ppr4Δ* and ETC mutants (Figs. 3C, 5A, EV3C and EV4A,B). It is likely that additional transcription factors and pathways contribute to the upregulation of stress response genes in response to mitochondrial dysfunction. A role for Pap1 downstream of mitochondrial dysfunction induced ROS is also indicated by *pap1Δ* cells behaving similarly to wild-type in a liquid TTC assay (Fig. EV2A).

Consistent with nuclear Pap1 being required to induce *caf5⁺* in *cup1-tt* and *ppr4Δ* cells, the upregulation of *caf5⁺* gene expression in both mutants appears to be Pap1-dependent (Fig. 5D). If the resistance phenotypes of *cup1-tt* and *ppr4Δ* cells are dependent on Pap1 and activation of the COSR pathway, deletion of the *pap1⁺* gene in these mutants would be predicted to abolish caffeine resistance. Indeed, *pap1Δ cup1-tt* and *pap1Δ ppr4Δ* double mutants exhibit reduced caffeine resistance, suggesting that resistance is Pap1-dependent (Fig. EV4D). However, this interpretation is

**A**

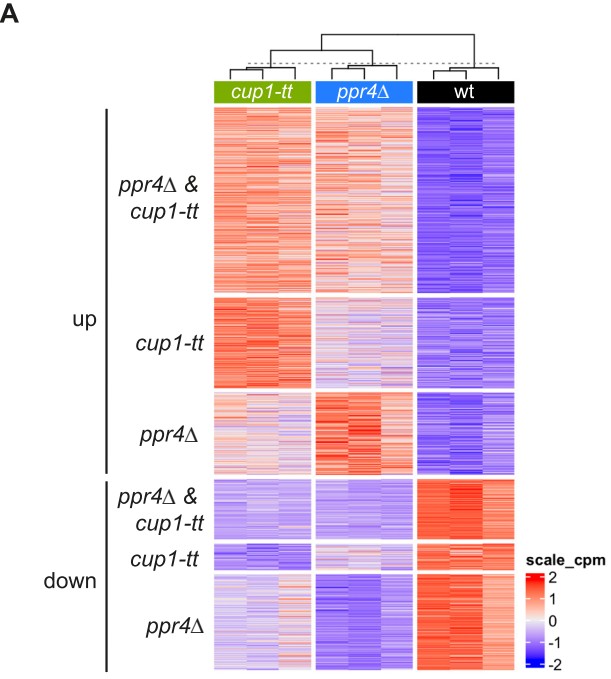

**B**

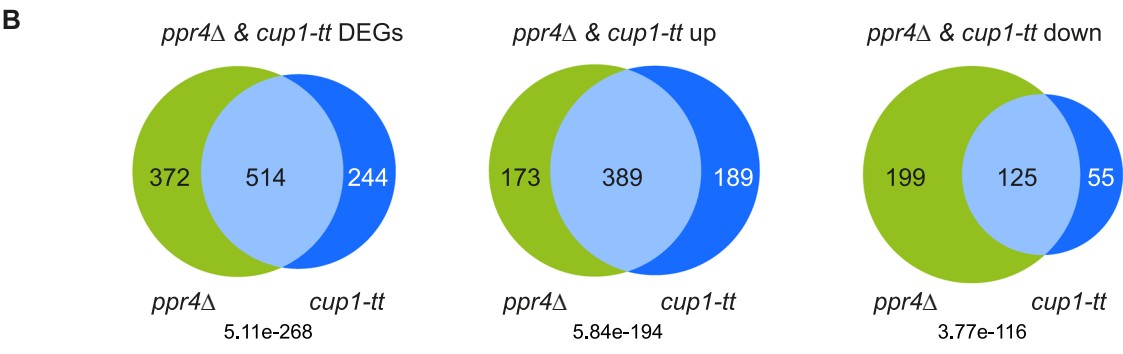

**C**

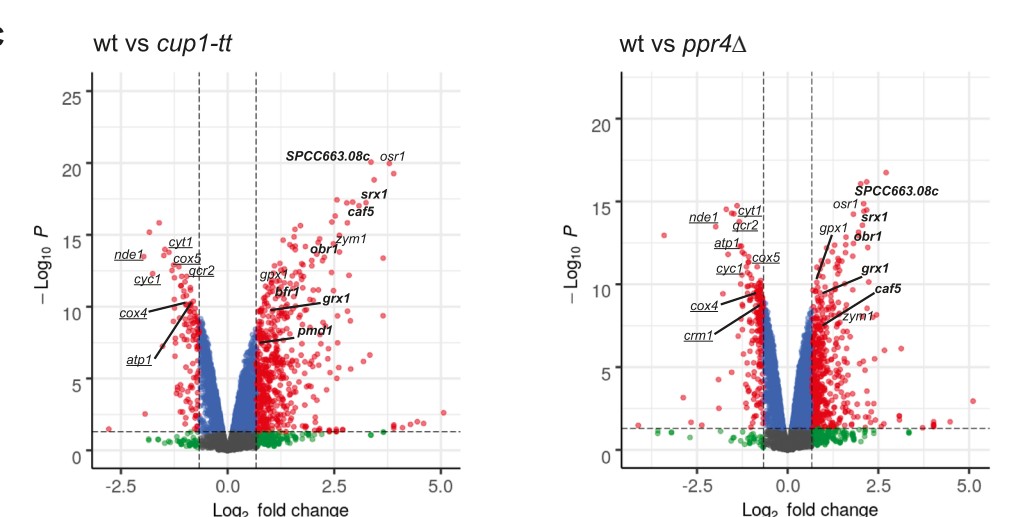

**Figure 3. RNA-Seq analyses reveal altered transcriptional programme in cells defective for Cup1 or Ppr4 function.**

(A) Heat-map of differentially expressed genes with fold-change value ≥ 1.5 and FDR-adjusted p-value < 0.05. Three biological replicates, one per column, for each genotype are shown and each row represents a gene (1130 unique genes plotted). Colour gradient key represents counts per million normalised per row (z-value), red for upregulated and blue for downregulated transcripts. Hierarchical clustering dendrogram displayed above the heatmap columns illustrates sample similarity based on gene expression profiles. Clustering was performed using Euclidean distance and complete linkage. Branch lengths represent similarity between samples, with closer branches indicating higher similarity. Dashed lines highlight the grouping of biological replicates within each condition. (B) Venn diagrams showing the differentially expressed genes in *cup1-tt* (dark blue) and *ppr4Δ* (green) cells and their overlap (light blue). Left: all differentially expressed genes (DEGs), middle: upregulated genes, and right: downregulated genes. P-values represent probabilities that the observed overlaps occurred by chance, as determined by hypergeometric test. (C) Volcano plot visualisation of transcriptional changes in *ppr4Δ* and *cup1-tt*. Volcano plots of fold change ($\log_2$-fold change) vs statistical significance ($-\log_{10}$ p-value) of RNA-seq data from *cup1-tt* (left panel) and *ppr4Δ* (right panel) vs wild-type. Genes that have p-value < 0.05 and fold change value ≥ 1.5 (dotted lines vertical and horizontal, respectively) are indicated by red and represent DEGs. Representative repressed genes are labelled from ETC complexes in GO:BP aerobic respiration (Table EV1) and induced genes from GO:BP detoxification (Table EV1). MFS-type transporter *caf5* is labelled. Genes under Pap1 control are in bold and nuclear-encoded ETC genes are underlined. Data plotted is from biological triplicates (n = 3 for each of wt, *ppr4Δ* and *cup1-tt*). Statistical significance determined using edgeR glmQLFTest. Source data are available online for this figure.

somewhat confounded by the fact that *pap1Δ* alone renders cells sensitive to external insults due to downregulation of Bfr1/Hba2 (Calvo et al, 2009; Liu et al, 2018).

Thus, the mitochondrial defects in *cup1-tt* or *ppr4Δ* cells, or cells lacking various ETC components, result in increased cytoplasmic ROS to levels sufficient to cause Pap1 to accumulate in nuclei where it can bind and activate downstream Pap1-dependent genes.

## Cells with compromised Cup1 or Ppr4 function display increased efflux

Mild oxidative stress activates the Pap1-dependent response and confers tolerance to caffeine through the upregulation of the genes encoding Bfr1 and Caf5 transmembrane transporters of the ABC and MFS families and is expected to enhance the export of toxic compounds from cells (Calvo et al, 2009). In comparison to wild-type, *cup1-tt* and *ppr4Δ* cells exhibit a measurable increase in *caf5+* gene expression (Figs. 3C, 5D and EV4B). In addition, *cup1-tt* cells induce expression of *bfr1+* (Fig. 3C). Efflux pumps such as transmembrane transporters reduce the intracellular concentration of compounds by expelling them from cells, thereby minimising their cytotoxicity (Engle and Kumar, 2024). The Rhodamine 6 G (R6G) fluorescent dye allows measurement of efflux due to transmembrane transporter activity (Gbelska et al, 2017). *cup1-tt*, *ppr4Δ* and some mutants with defective mitochondrial function (*qcr7Δ*, *cox4Δ* and *rpm1Δ*) exhibit a substantially higher rate of R6G efflux compared to wild-type or cells lacking the four main transmembrane transporters: *bfr1Δ caf5Δ mfs1Δ pmd1Δ* (Kawashima et al, 2012) (Figs. 5E and EV4E). Chlorgyline is a broad-spectrum inhibitor of Major Facilitator Superfamily (MFS) and ATP-Binding Cassette (ABC) fungal efflux pumps (Holmes et al, 2012). Addition of chlorgyline (CHL) to media suppressed the caffeine resistance of both *cup1-tt* and *ppr4Δ* and, at higher concentrations, *hba1Δ* cells (Fig. 5F).

The increased efflux observed in *cup1-tt* and *ppr4Δ* cells and cells lacking particular ETC components, and the suppression of caffeine resistance by an efflux pump inhibitor suggests that their resistance phenotype results from the Pap1-mediated upregulation of genes encoding transmembrane transporters.

## Caffeine-resistant epimutants display mitochondrial dysfunction and increased efflux

Phenotypes of both *cup1-tt* and *ppr4Δ* cells are consistent with these mutations causing mitochondrial dysfunction that leads to

increased ROS levels which activates the MNR and Pap1-dependent oxidative stress response pathways, in turn causing upregulation of transmembrane transporters and increased efflux of toxic compounds. To determine whether the resistance phenotypes of epimutants to such insults is due to the same mechanisms demonstrated for the stable *cup1-tt* and *ppr4Δ* genetic mutants, cells harbouring caffeine-resistant epimutations were tested in several assays to evaluate key aspects of the proposed mechanism of resistance.

Epimutants *hba1-epi1*, *cup1-epi20* and *ppr4-epi3* (previously named UR1, UR20 and UR3, respectively) all displayed resistance to caffeine (Fig. 6A), as expected (Torres-Garcia et al, 2020b). They each also showed resistance to both clinical and agricultural azoles similar to their respective mutants (Fig. 6A). Both *cup1* and *ppr4* epimutants show reduced growth on non-glucose carbon sources compared to wild-type, and a mixture of red and white colonies in a TTC-overlay assay. The *cup1-epi20* and *ppr4-epi3* epimutants displayed similar phenotypes to their cognate *cup1-tt* and *ppr4Δ* genetic mutants (Figs. 6A and EV5A), but *cup1-epi20* and *ppr4-epi3* (UR3) epimutant growth on various media is consistent with a mixed population of wild-type-like and mutant-like phenotypes. For instance, homogeneous genetic mutants do not grow (*ppr4Δ*) or exhibit uniform slow growth (*cup1-tt*) on media in which glycerol is the sole carbon source, whereas *cup1-epi20* and *ppr4-epi3* epimutants exhibit different levels of heterogeneity with only a proportion of cells forming colonies on this glucose-deficient media (Fig. 6A). In contrast, *hba1-epi1*, that is not expected to affect mitochondrial function, grows similarly on non-selective media containing either glucose or glycerol and does not show variegating colour in the TTC assay. Growth assays of other caffeine-resistant epimutants reveal that UR4 is strongly resistant to azole-based antifungals, whereas UR6 shows weaker resistance to these drugs, while UR5 is sensitive (Fig. 6B). These three epimutants all produced mainly red colour in the TTC assay and grew well on non-glucose carbon sources (Fig. EV5A), suggesting that they do not have dysfunctional mitochondria. These epimutants showed ROS levels similar to those of wild-type (Fig. 6C), suggesting they utilize a different, ROS-independent, mechanism to develop resistance.

Consistent with their TTC behaviour, cytology and flow cytometry profiles of *cup1* and *ppr4* epimutants stained with the DCFH-DA ROS sensor are also suggestive of mixed populations: epimutants exhibit a biphasic DCF fluorescence profile, with subpopulations of cells exhibiting either wild-type or mutant-like ROS levels (Figs. 6C and EV5B). Similar to *cup1-tt* and *ppr4Δ*

## A

### Retrograde response

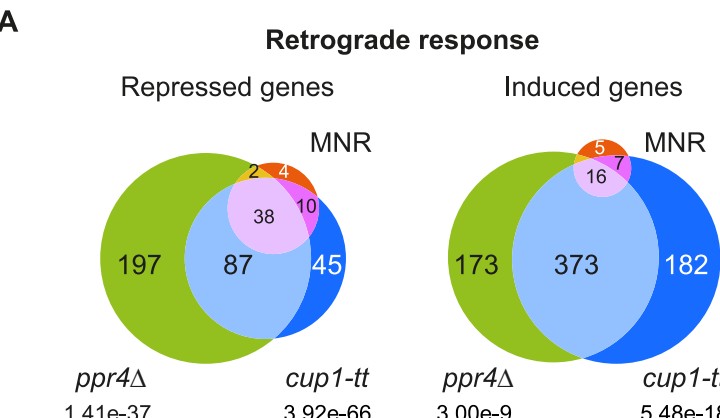

### Repressed genes

MNR

| | |
|---|---|
| 2 | 4 |
| 38 | 10 |

197 87 45

*ppr4Δ* *cup1-tt*

1.41e-37 3.92e-66

### Induced genes

MNR

5
16 7

173 373 182

*ppr4Δ* *cup1-tt*

3.00e-9 5.48e-18

## B

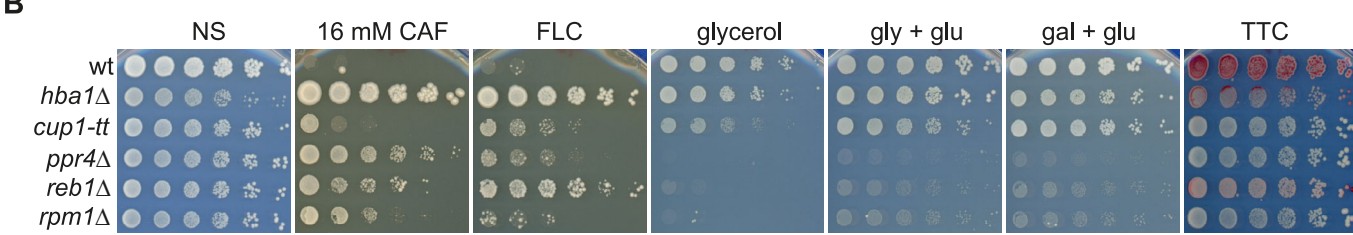

| | NS | 16 mM CAF | FLC | glycerol | gly + glu | gal + glu | TTC |
|---|---|---|---|---|---|---|---|
| wt | | | | | | | |
| *hba1Δ* | | | | | | | |
| *cup1-tt* | | | | | | | |
| *ppr4Δ* | | | | | | | |
| *reb1Δ* | | | | | | | |
| *rpm1Δ* | | | | | | | |

## C

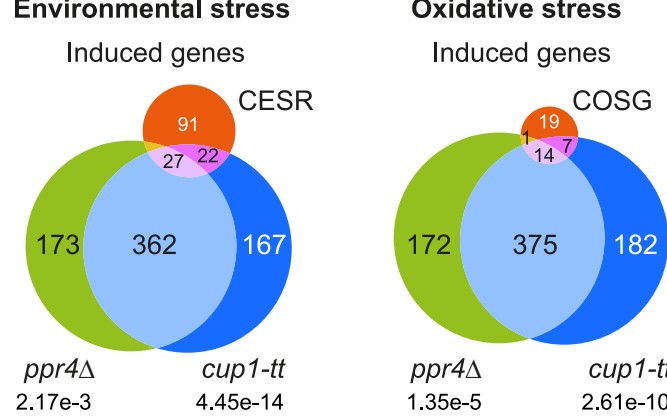

### Environmental stress

Induced genes

CESR

91
27 22

173 362 167

*ppr4Δ* *cup1-tt*

2.17e-3 4.45e-14

### Oxidative stress

Induced genes

COSG

19
1 14 7

172 375 182

*ppr4Δ* *cup1-tt*

1.35e-5 2.61e-10

**Figure 4. Transcriptional profiles of cells defective for Cup1 or Ppr4 overlap with mitonuclear retrograde, core oxidative stress genes and core environmental stress response pathways.**

(**A**) Comparison of *cup1-tt* and *ppr4Δ* with mitonuclear retrograde response (Malecki et al, 2016). Venn diagrams comparing genes repressed (left) or induced (right) in *cup1-tt* (dark blue), *ppr4Δ* (green) and in the mitonuclear retrograde response (MNR; pink). *p*-values represent probabilities that the observed overlaps occurred by chance, as determined by hypergeometric test. (**B**) Growth assay to assess mitochondrial function and resistance to insults. Five-fold serial dilutions of indicated strains spotted onto non-selective plates (NS; YES media) or plates containing 16 mM caffeine (CAF) or 0.3 mM fluconazole (FLC), or YES plates in which glucose was replaced with 3% glycerol or 3% glycerol + 0.1% glucose (gly + glu) or 2% galactose + 0.1% glucose (gal + glu). Plates photographed after 2–8 days at 32 °C. After 3 days' growth, colonies on YES were overlaid with TTC-containing agarose and incubated for ~24 h to assess respiratory competence. (**C**) Comparison of *cup1-tt* and *ppr4Δ* with: left, core environmental stress response (CESR) (Chen et al, 2003) and right, core oxidative stress genes (COSG) (Chen et al, 2008). Venn diagrams comparing genes upregulated in *cup1-tt* (dark blue) and *ppr4Δ* (green) and core genes induced in Left, core environmental stress response (CESR; pink) and Right, core oxidative stress genes (COSG; pink). *p*-values as in (**A**). See also Table EV1. Source data are available online for this figure.

mutants, *cup1-epi20* and *ppr4-epi3* epimutants display increased efflux of R6G compared to wild-type (Fig. 6D). As with *cup1-tt* and *ppr4Δ*, the resistant phenotypes of all epimutants could also be counteracted by addition of the efflux pump inhibitor chlorgyline to the media (Fig. 6E).

These data demonstrate that the *cup1* and *ppr4* caffeine-resistant epimutants exhibit a set of phenotypes similar to those of their genetic *cup1-tt* and *ppr4Δ* counterparts. The results are consistent with heterochromatin-mediated repression of the underlying wild-type *cup1+* or *ppr4+* mitochondrial protein-encoding genes in the

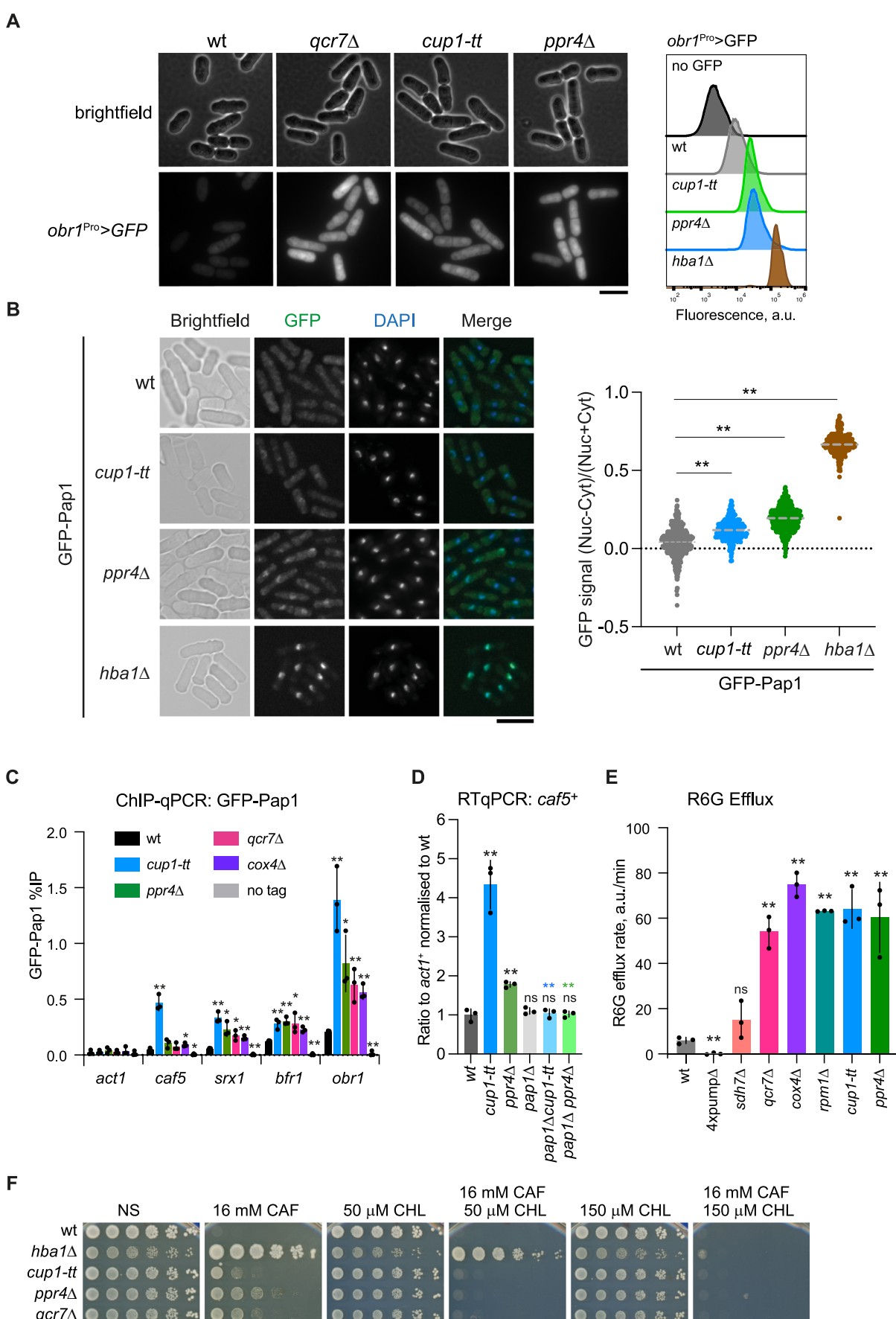

◄ **Figure 5. Cup1 and Ppr4 deficient cells activate the Pap1-dependent stress response and exhibit increased efflux.**

(A) Pap1-dependent *obr1*-promoter reporter assay. Fluorescence microscopy images (left) and flow cytometry (right) of live cells of indicated strains containing GFP under control of the Pap1-dependent *obr1* promoter. See also Fig. EV4A. Fluorescence images are scaled relative to the brightest image. Scale bar, 10 μm. (B) Localisation of GFP-Pap1. Left: Fluorescence microscopy of fixed cells of indicated strains containing GFP-Pap1 and stained with DAPI. Right: quantification of images. The ratio of nuclear minus cytoplasmic signal (Nuc–Cyt) to total (nuclear plus cytoplasmic signal; Nuc+Cyt) is plotted. Cells analysed for each strain, from left to right, $n = 416, 289, 451, 355$. Dotted line indicates median. Significance of the difference between samples was evaluated by Kruskal–Wallis test followed by Dunn's multiple comparison post-hoc test. Asterisks indicate significance, $**p < 0.01$. (C) ChIP-qPCR to assess GFP-Pap1 binding to Pap1-dependent promoters. Promoter regions of the *caf5* and *bfr1* (transmembrane transporters), *srx1* (sulfiredoxin), and *obr1* (dehydrogenase) genes were analysed by qPCR to determine GFP-Pap1 % immunoprecipitation in the indicated strains. *act1* serves as negative control locus. Data are mean and standard deviation from three biological replicates. *p* values for mutants compared to wt determined by two-tailed Student's t-test: $*p < 0.05$; $**p < 0.01$; unmarked columns are not significant. (D) Dependence of *caf5* transcript levels on Pap1. Quantification by RT-qPCR of transcript levels of the plasma membrane transporter Caf5 in the indicated strains. Data are mean and standard deviation from three biological replicates. *p* values determined by two-tailed Student's t-test: $*p < 0.05$; $**p < 0.01$; ns, not significant. Black stars, comparison with wt; blue stars, *cup1-tt* vs *cup1-tt pap1Δ*; green stars, *ppr4Δ* vs *ppr4Δ pap1Δ*. (E) Efflux of Rhodamine 6G (R6G) from cells. Cells of the indicated strains were preloaded with R6G and R6G release to the media measured upon addition of glucose (as YES). Rates of efflux expressed as arbitrary units/min. See also Fig. EV4E. *4xpumpΔ* strain four transmembrane transporters are absent (Bfr1, Pmp1, Caf5, Mfs1) along with Pap1 and Prt1. Data are mean and standard deviation from three biological replicates. *p* values determined by two-tailed Student's t-test: $*p < 0.05$; $**p < 0.01$; ns, not significant. (F) Growth assay to assess impact of efflux pump inhibitor chlorgyline on caffeine resistance. Five-fold serial dilutions of indicated strains spotted onto non-selective plates (NS; YES media) or plates containing 16 mM caffeine and/or the efflux pump inhibitor chlorgyline (CHL) at the indicated concentrations. Plates photographed after 2–8 days at 32 °C. *P* values (left to right): (B) all <0.0001. (C) *act1*: 0.8149, 0.3759, 0.9029, 0.5742, 0.6395; *caf5*: 0.0005, 0.0781, 0.2355, 0.0240, 0.0118. *srx1*: 0.0007, 0.01, 0.016, 0.0014, 0.0011. *bfr1*: 0.0044, 0.0010, 0.0401, 0.0023, <0.0001. *obr1*: 0.0022, 0.0138, 0.0081, 0.0009, 0.0003. (D) mutants vs wt: 0.0009, 0.0015, 0.3872, 0.8472, 0.7047. *pap1Δ cup1-tt* vs *cup1-tt*: 0.0009. *pap1Δ ppr4Δ* vs *ppr4*: 0.0003. (E) 0.0022, 0.1384, 0.0003, <0.0001, <0.0001, 0.0003, 0.0039. Source data are available online for this figure.

*cup1* and *ppr4* epimutants as being the main contributors to elevated intracellular ROS levels that drive increased efflux.

## Variegation within epimutant populations permits phenotypic plasticity

The limited growth of epimutants on glycerol and the mix of TTC-white respiration-defective and TTC-red respiration-competent colonies is suggestive of at least two subpopulations of cells. To explore this apparent heterogeneity of epimutants further, colonies exhibiting distinct phenotypes were isolated from *cup1-epi20* and *cup1-epi28* (originally UR28; (Torres-Garcia et al, 2020b)) epimutant populations (Fig. 7A). TTC-white colonies (respiration-defective; *cup1-epi20w, cup1-epi28w*) and colonies growing on non-fermentable carbon sources (respiration competent; *cup1-epi20g, cup1-epi28g*) were picked and re-tested in serial dilution assays. Like *cup1-epi20* and *cup1-epi28*, their respective TTC-white derivatives could form resistant colonies on caffeine and fluconazole plates, but *cup1-20w* and *cup1-28w* did not grow at all on glycerol plates (Figs. 7B and EV6A). In contrast, the respiration-competent *cup1-epi20g* and *cup1-epi28g* derivatives grew well on glycerol plates but were sensitive to both caffeine and fluconazole (Figs. 7B; and EV6A). Derivatives of epimutants obtained by extensive growth on non-selective YES media, *cup1-epi20y* and *cup1-epi28y*, also grew well on non-glucose-containing media and were sensitive to insults, similar to wild-type cells (Figs. 7B and EV6A). Flow cytometry analysis with the ROS sensor DCFH-DA revealed that ROS levels correlated with respiration competence: TTC-white derivative *cup1-epi20w* exhibited high ROS levels whereas derivatives selected on non-fermentable carbon sources, *cup1-epi20g*, or extensive non-selective growth, *cup1-epi20y*, had low ROS levels (Fig. 7C). A derivative that had partially lost resistance, *cup1-epi20yp* (Fig. EV6B), showed a partial shift in ROS levels with approximately half the cells showing mutant-like and half wild-type-like ROS levels (Fig. 7C). Moreover, consistent with mitochondrial dysfunction-dependent high cellular ROS driving efflux, the respiration-defective *cup1-epi20* and *cup1-epi20w* derivative displayed higher efflux than the respiration-competent

*cup1-epi20g and cup1-epi20y* derivatives (Fig. 7D). Thus, epimutants exhibit variegation where the ratio of high ROS/insult-resistant to low ROS/insult-sensitive cells fluctuates in a cell population, and different states can be selected depending on the growth environment. Not all *cup1* epimutants behaved like *cup1-epi20* and *cup1-epi28*; for example, *cup1-epi15* only partially lost caffeine resistance upon selection on non-fermentable carbon sources (Fig. EV6C), suggesting that additional changes may arise in some initially unstable resistant epimutants that alter their subsequent stability and responsiveness (Torres-Garcia et al, 2020b).

These analyses indicate that epimutants generated by heterochromatin island formation over the *cup1+* or *ppr4+* genes impose mitochondrial dysfunction which leads to increased efflux and hence resistance of otherwise wild-type *S. pombe* cells to both caffeine and azole antifungal drugs. Thus, in this model fungal system, cells can adapt to external insults by altering their metabolic competence to reduce intracellular levels of toxic compounds, allowing survival in otherwise adverse conditions.

## Discussion

Two unstable caffeine-resistant epimutants associated with H3K9me heterochromatin islands have now been shown to contain genes encoding mitochondrial proteins: *cup1+* and *ppr4+*. Genetic mutations at either locus exhibit caffeine and azole-based antifungal resistance similar to those observed in the cognate epimutants. Transcriptome profiling revealed that the mito-nuclear retrograde response (MNR), core environmental stress response (CESR) and the core oxidative stress gene (COSG) pathways were activated in both *cup1-tt* and *ppr4Δ* mutants. Elevated ROS levels were detected in *cup1-tt* and *ppr4Δ* mutants and the original *cup1-epi* and *ppr4-epi* epimutants indicating that the resulting mitochondrial dysfunction generates a source of oxidative stress. *cup1* and *ppr4* genetic mutants and epimutants exhibited increased efflux rates, consistent with higher intracellular ROS triggering Pap1 nuclear retention where it drives increased expression of COSG

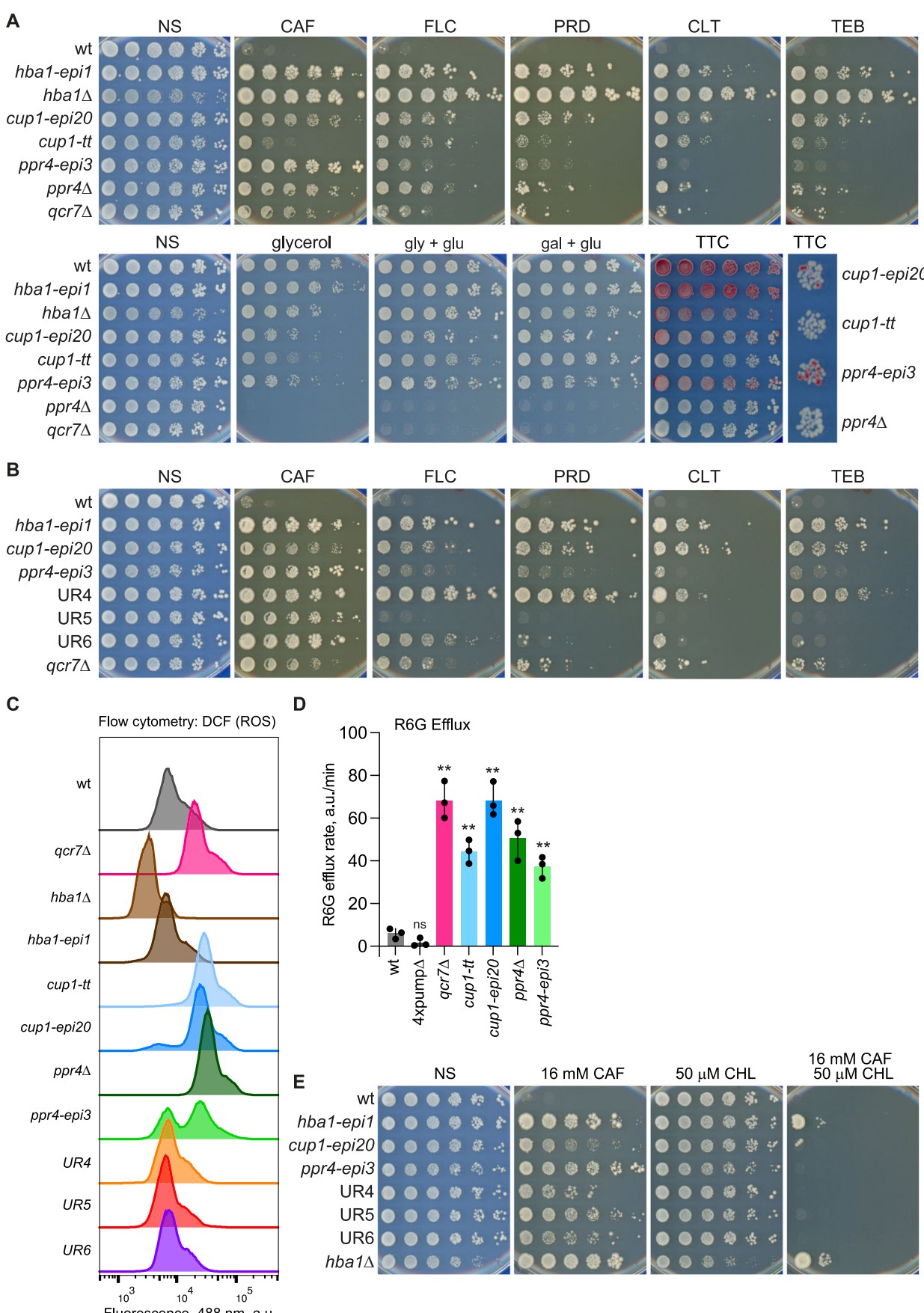

**Figure 6.** *cup1* and *ppr4* epimutants show respiratory deficiency, increased ROS and enhanced efflux.

(A) Growth assay to assess mitochondrial competence and resistance to insults of *hba1*, *cup1* and *ppr4* mutants and epimutants. Upper: Five-fold serial dilutions of indicated strains spotted onto non-selective plates (NS; YES media), or YES plates in which glucose was replaced with 3% glycerol or 3% glycerol + 0.1% glucose (gly + glu) or 2% galactose + 0.1% glucose (gal + glu). Plates photographed after 2–8 days at 32 °C. TTC: after 3 days' growth, colonies on YES were overlaid with TTC-containing agarose and incubated for ~24 h to assess respiratory competence. Lower: Five-fold serial dilutions of indicated strains spotted onto non-selective plates (NS; YES media) or plates containing 16 mM caffeine (CAF), or azole-based antifungals: 0.3 mM fluconazole (FLC), 0.4 µg/ml prothioconazole-desthio (PRD), 50 µg/ml clotrimazole (CLT), 1.6 µM tebuconazole (TEB). Plates photographed after 2–8 days at 32 °C. Right panel: zoom-in of colonies of indicated strains. (B) Growth assay to assess resistance to caffeine and azole antifungals. Five-fold serial dilutions of indicated epimutant strains spotted onto plates as in (A). (C) ROS levels in epimutants. Flow cytometry of cells of the indicated strains stained with 10 µM DCFH-DA and analysed by flow cytometry at 488 nm. (D) Rates of R6G efflux of the indicated strains. Data are mean and standard deviation from three biological replicates. *P* values determined by two-tailed Student's t-test: *$p < 0.05$; **$p < 0.01$; ns, not significant. (E) Impact of the efflux pump inhibitor chlorgyline on caffeine resistance of epimutant strains. Five-fold serial dilutions of indicated strains spotted onto non-selective plates (NS; YES media) or plates containing 16 mM caffeine and/or the efflux pump inhibitor chlorgyline (CHL) at 50 µM. Plates photographed after 2–8 days at 32 °C. *P* values (left to right): (D) 0.0761, 0.0003, 0.0004, 0.0002, 0.0014, 0.0006. Source data are available online for this figure.

genes including efflux pumps and consequently resistance to external insults such as caffeine and azole-based antifungals (model, Fig. 7E). Epimutations at two separate loci impose mitochondrial dysfunction in wild-type cells. Thus, intrinsically generated ROS triggers the oxidative stress that ultimately leads to caffeine and antifungal resistance. Likewise, inhibition of complex III with Antimycin A conferred resistance, whereas overexpression of a hydrogen peroxide scavenger protein mildly reduced resistance. The specific gene or genes whose repression mediates resistance phenotypes in the UR4, UR5 and UR6 epimutants remain to be identified. Their wild-type-like ROS levels, TTC behaviour and growth on non-glucose carbon sources suggest that the cause of resistance in these epimutants does not primarily involve mitochondrial dysfunction. Nonetheless, three epimutation loci impact genes which impinge on the Pap1-dependent stress response pathway: *hba1⁺* (UR1), *cup1⁺* (UR2, UR8 to 30) and *ppr4⁺* (UR3, UR7), suggesting that genes in this pathway are a fruitful target for heterochromatin-mediated repression to deliver transient resistance to insults.

Mechanisms resulting in caffeine resistance have been well characterised in fission yeast. Initial studies mainly identified components affecting the Pap1 pathway (Benko et al, 1998; Kumada et al, 1996; Papadakis and Workman, 2015; Yaakoub et al, 2022). Subsequently, caffeine stress was found to induce the Sty1 MAPK branch, rather than the Pap1 branch of the oxidative stress response. It is known that Sty1 hyperactivation does not result in caffeine resistance, instead, aberrant activation of the Pap1 pathway has been shown to result in caffeine resistance, primarily through enhanced toxin export from cells via Pap1-dependent efflux pump upregulation (Calvo et al, 2009). Mutations that prevent Pap1 export from the nucleus (*hba1Δ*) or that cause a general increase in Pap1 levels (e.g. defective proteasome-mediated degradation) exhibit multidrug-resistant phenotypes (Marte et al, 2020; Penney et al, 2012). Even a modest increase in the overall level of Pap1 is sufficient to increase tolerance of *S. pombe* to external insults. Any mutation or epimutation that causes an increase in Pap1 levels or its oxidation-mediated retention by nuclei is expected to confer a resistance phenotype (Benko et al, 1998; Calvo et al, 2012, 2009; Castillo et al, 2003; Kumada et al, 1996). Indeed, although recruitment of GFP-Pap1 to Pap1-dependent genes such as the transmembrane transporters *caf5⁺* and *bfr1⁺* is modest in *cup1-tt* and *ppr4Δ* cells compared to treatment with exogenous hydrogen peroxide (Fig. EV4C), efflux rates are elevated. It is likely that other stress-sensing pathways and transcription factors also contribute to transporter upregulation

and increased efflux in response to mitochondrial dysfunction. *S. pombe* genetic mutants lacking mitochondrial proteins are known to accumulate ROS and some, such as *phb2Δ*, exhibit Pap1-dependent resistance to external insults (Liu et al, 2018; Zuin et al, 2008). Like *S. pombe* Pap1, *S. cerevisiae* oxidation sensitive transcription factor Yap1 accumulates in nuclei in response oxidative stress and activate genes encoding drug efflux ABC transporters such as Pdr5 (Buechel and Pinkett, 2020; Morgan and Veal, 2007). Moreover, most human (*Aspergillus, Candida, Cryptococcus*) and cereal crop (*Magnaporthe, Ustilago, Zymoseptoria*) fungal pathogens utilise AP-1-like/Yap1-related proteins to mitigate oxidative stress and external insults (Yaakoub et al, 2022).

Mitochondrial dysfunction is known to trigger pleiotropic resistance pathways, including azole-based antifungal agents and other compounds, in many fungal species including pathogens (Dhar et al, 2019; Gale et al, 2023; Peng et al, 2012; Shingu-Vazquez and Traven, 2011; Song et al, 2020). Indeed, comprehensive analyses of *S. cerevisiae* wild-type populations have demonstrated cell-to-cell variation in proliferation rates. Approximately 10% of cells exhibit slow growth correlated with defective mitochondrial function resulting from low mitochondrial genome (mtDNA) copy number. Moreover, these slow-growing cells exhibited increased resistance to fluconazole due to Pdr5 efflux pump upregulation (Dhar et al, 2019). Thus, in *S. cerevisiae*, natural fluctuations in mtDNA copy number, is apparently one factor underlying heterogeneity in mitochondrial function and associated antifungal resistance. *S. cerevisiae* and some other yeasts can completely lose mtDNA, forming petite colonies. *S. pombe* is known to be unable to completely lose mtDNA (i.e. petite-negative) and it was recently shown that *S. pombe* cells struggle in anaerobic environments associated with fermentation due to ongoing reliance on respiration to generate particular amino acids (Malecki et al, 2020). A single nucleotide polymorphism in *pyk1* reduces pyruvate kinase activity and alters the metabolic balance towards respiration over fermentation in fission yeast. This results in increased stress tolerance at the cost of growth demonstrating an adaptive trade-off in metabolic regulation (Kamrad et al, 2020). Mitochondrial dysfunction can be exploited by cells as a means of acquiring resistance to exogenous insults. Epimutations that affect respiratory competence in *S. pombe* provide an alternative route for cell-to-cell variation in the mode of metabolism engaged for proliferation. Indeed, both heterochromatin-mediated epimutations over nuclear-encoded mitochondrial genes and DNA-based heterogeneity in the form of mitochondrial heteroplasmy (Van Leeuwen

**A**

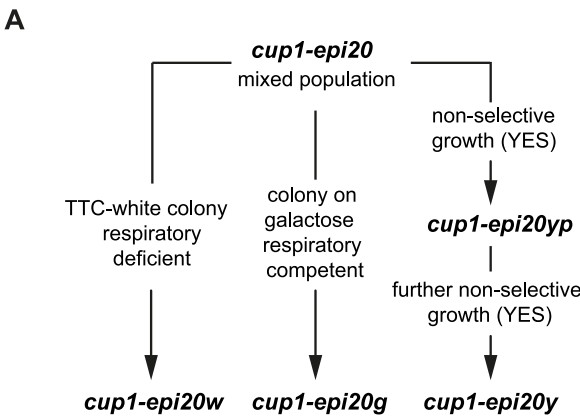

**B**

| | NS | CAF | FLC | glycerol | TTC | TTC |
|---|---|---|---|---|---|---|
| qcr7Δ | | | | | | -epi20 |
| wt | | | | | | |
| cup1-epi20 | | | | | | -epi20w |
| cup1-epi20w | | | | | | |
| cup1-epi20g | | | | | | -epi20g |
| cup1-epi20y | | | | | | |
| cup1-tt | | | | | | -epi20y |
| wt | | | | | | |

**C**

Flow cytometry: DCF (ROS)

wt
qcr7Δ
cup1-tt
epi20w
epi20
epi20yp
epi20g
epi20y

cup1 epimutant derivatives

Fluorescence, 488 nm, a.u

$10^3$  $10^4$  $10^5$

**D**

R6G Efflux

R6G efflux rate, a.u./min

wt
qcr7Δ — **
cup1-epi20 — **
cup1-epi20w — **
cup1-epi20g — ns
cup1-epi20y — ns

**E**

cup1 or ppr4 mutation or epimutation | Mitochondrial dysfunction ROS ↑ | COSR: Pap1 → nucleus | Resistance genes ↑

ETC
ROS
Pap1
MNR
Enhanced drug efflux

◄

**Figure 7. Selection of *cup1* epimutant subpopulations.**

(A) Isolation of *cup1-epi20* derivatives. *cup1-epi20w* derivative identified as TTC-white colony; *cup1-epi20g* derivative selected for respiratory competence (growth on galactose + 0.1% glucose); *cup1-epi20yp* derivative isolated after extensive growth on non-selective YES media; *cup1-epi20y* derivative isolated after further growth on non-selective YES media; further details in Methods. (B) Growth assay to assess mitochondrial competence and resistance to insults of *cup1* epimutant derivatives described in (A). Panels show five-fold serial dilutions of indicated isolates spotted onto non-selective plates (NS; YES media) or plates containing 16 mM caffeine, 0.3 mM fluconazole (FLC), YES containing 3% glycerol rather than glucose, and TTC overlay to assay respiratory competence. Right panel: zoom-in of colonies of indicated epimutant derivatives. (C) ROS levels in epimutant derivatives. Flow cytometry (488 nm) of DCFH-DA-stained cells of the *cup1-epi20* derivatives described in (A), and a derivative which had partially lost resistance (*cup1-epi20yp*; see Fig. EV6C) along with wild-type, *cup1-tt* and *qcr7Δ* cells. (D) Efflux in epimutant derivatives. R6G efflux rate of the indicated *cup1-epi20* derivatives described in (A), along with wild-type and *qcr7Δ* strains as controls. Rates expressed as arbitrary units/min. Data are mean and standard deviation from four biological replicates. *p* values determined by two-tailed Student's t-test: *$p < 0.05$; **$p < 0.01$; ns, not significant. (E) Model: Mutation or heterochromatin-mediated repression of *cup1+* or *ppr4+* causes mitochondrial dysfunction due to defective electron transport chain (ETC) which causes increased levels of reactive oxygen species (ROS; red stars). ROS mildly activates the Pap1-dependent oxidative stress response, causing increased levels of Pap1 in nuclei (red arrow) and increased binding at Pap1-dependent promoters. Genes encoding antioxidants and drug-resistance proteins are upregulated, including transmembrane transporters that lead to increased efflux of insults such as caffeine. The mitonuclear retrograde response is also activated leading to decreased expression of nuclear-encoded mitochondrial proteins (black arrow). *P* values (left to right): (D) 0.0003, 0.0006, <0.0001, 0.4230, 0.2428. Source data are available online for this figure.

et al, 2008) could be harnessed to provide phenotypic plasticity advantageous in changing environments.

Components of the oxidative phosphorylation pathway are expressed from both the nuclear and mitochondrial genomes. Mito-nuclear communication and co-ordination of expression is crucial to ensure balanced mitochondrial protein expression (Liu and Butow, 2006). The MNR pathway signals from mitochondria to the nucleus to modulate transcription and ensure co-ordinated expression of nuclear and mitochondrial genes encoding mitochondrial components (Kotiadis et al, 2014). In addition to inducing a mild oxidative stress response, the *cup1-tt* and *ppr4Δ* mutants activate the MNR pathway, resulting in the reduced expression of many nuclear genes that encode mitochondrial proteins, including numerous ETC components (Figs. 3C, 4 and EV3). It is likely that cells in *cup1* and *ppr4* epimutant populations with defective mitochondrial function also activate the MNR pathway, resulting in repression of ETC component genes. The regulators and details of how the MNR operates in *S. pombe* remain unknown. However, ultimately transcriptional repression of nuclear-encoded ETC genes may protect cells by reducing ROS accumulation and associated oxidative damage (Malecki et al, 2016).

An optimum balance between fermentation and respiration is required to cater for the energy and metabolic requirements of cells. When glucose is abundant, Crabtree-positive fungi—such as *S. pombe*—utilise glucose inefficiently through fermentation (Malecki et al, 2016). When glucose is limited *S. pombe* cells switch to respiration-dependent growth and proliferate more slowly on non-fermentable carbon sources such as glycerol. ETC components such as Qcr7 and Rip1 are required for oxidative phosphorylation and efficient respiration (Alam et al, 2023; Malecki et al, 2016). In ETC mutants, mitochondrial function is defective, meaning growth is not supported on non-glucose carbon sources, but aerobic glycolysis and/or fermentation enables near-normal growth in glucose-rich medium.

Unlike wild-type cells or their genetic mutant counterparts, *cup1* epimutants display variegation of the resistance phenotype and their ability to grow on different carbon sources. Growing *cup1-epi* cells on non-glucose carbon sources selected for respiration-competent cells in the epimutant population which had normal levels of ROS and efflux, resulting in caffeine and azole sensitivity. Conversely, isolating TTC-white colonies selected respiration-defective *cup1-epi* derivatives which were caffeine resistant and displayed high levels of ROS and efflux. Thus, once established in a population, epimutations can provide a source of phenotypic heterogeneity, and the environmental context (non-glucose rather

than glucose-rich media) can influence the relative proportions of sensitive and resistant cells by promoting growth of respiration competent cells with normal mitochondrial function. The essential nature of the *cup1+* gene is also likely to influence the degree to which heterochromatin can spread and repress its transcription before deleterious effects on cell fitness occur and will affect the balance between mutant-like and wild-type phenotypes in the population. Respiration-competent revertants of some epimutants (e.g. *cup1-epi15*; Fig. EV6C) lost azole resistance but only partially lost caffeine resistance, suggesting additional alterations may bolster resistance. Previous analyses indicated that several caffeine-resistant epimutants accumulated extra-chromosomal circles (Torres-Garcia et al, 2020b). Indeed, their instability makes epimutants inherently difficult to work with. Off selection they can lose the primary features responsible for resistance, whilst continued selection inevitably results in additional changes that can stabilise the resistant phenotype. Thus, epimutations may act as a gateway, or may act in conjunction with DNA-based changes or other mechanisms to confer robust resistance. Nonetheless, epimutations allow the level of mitochondrial dysfunction to be tuned so that, even for essential genes such as *cup1+*, the fitness cost can be balanced with survival in the presence of the external insult.

Resistance to antifungal drugs is rising in clinical and agriculture settings (Fisher et al, 2022). Fungal infections cause over 1.5 million human deaths annually (Bongomin et al, 2017). Food security is also threatened by plant fungal pathogens which cause crop losses of up to 20% (Fones et al, 2020). Considering that ETC defects lead to increased efflux and drug resistance, it is possible that crop-spraying with ETC inhibitors (i.e. QoI and QiI complex III-specific compounds) (Sierotzki, 2015) could counterproductively contribute to increased resistance to azole-based antifungals. It is likely that heterochromatin-dependent epimutations could contribute to antifungal resistance in clinical and agricultural fungal pathogens. A form of unstable resistance to antifungal drugs, termed heteroresistance, has been observed in human pathogenic fungi such as *Candida albicans* and *Cryptococcus neoformans* (Berman and Krysan, 2020; Stone et al, 2019). Moreover, unstable resistance, reminiscent of heteroresistance, is apparent in the major wheat pathogen *Zymoseptoria tritici* (Fouché et al, 2022; Gutiérrez-Alonso et al, 2017). Although such unstable resistance might result from aneuploidy or mitochondrial heteroplasmy, another plausible explanation is that epimutations are involved. Chromatin readers, writers and erasers identified in human fungal pathogens could

contribute to the formation of resistance-inducing epimutations in these organisms (Buscaino, 2019). RNAi-mediated epimutants have also been described which render the emerging human fungal pathogen *Mucor circinelloides* resistant to FK506 and rapamycin by reducing expression of the *fkbA* gene encoding the FKBP12 target protein (Calo et al, 2014; Pérez-Arques et al, 2020). Heterochromatin does not appear to be involved in forming these *M. circinelloides* FK506-resistant epimutants and resistance is not linked to mitochondrial dysfunction. However, it is likely that other fungal pathogens can adapt to the application of antifungal agents through heterochromatin- and/or RNAi-mediated epimutations that impair mitochondrial function thereby elevating intracellular ROS, activating Yap1-dependent efflux pumps to increase the export of the toxic compound. Heterochromatin-based epimutations that enable FK506 resistance have recently been reported in Mucor species (bioRxiv preprint https://doi.org/10.1101/2025.06.17.660219).

The analyses presented here demonstrate that mitochondrial dysfunction in fission yeast can lead to the accumulation of intracellular ROS which upregulates efflux rates via the canonical oxidative stress response. Epimutations are advantageous in fluctuating environments as they can be propagated but are unstable over many generations. To ensure survival in the presence of an insult, cells with an epimutation affecting mitochondrial function need to balance the damage that results from excessive ROS with the advantage resulting from ROS-mediated efflux. Hence, epimutations are ideal in that they allow cell-to-cell variation in the degree of mitochondrial dysfunction, resulting in efflux activity and associated resistance.

# Methods

### Reagents and tools table

| Reagent/Resource | Reference or Source | Identifier or Catalog Number |
| --- | --- | --- |
| **Experimental models** | | |
| Fission yeast *Schizosaccharomyces pombe* | See Strain List, Table EV2 | |
| **Recombinant DNA** | | |
| SpEDIT plasmids | Torres-Garcia et al, 2020a | |
| Bähler tagging plasmids | Bähler et al, 1998 | |
| **Antibodies** | | |
| Rabbit anti-GFP | Thermo Fisher Scientific (Life Technologies) | A-11122 |
| Peroxidase-coupled Goat anti-Rabbit secondary | Merck | A6154 |
| **Oligonucleotides and other sequence-based reagents** | | |
| | See Primer list, Table EV3 Primers obtained from Merck or Integrated DNA Technologies | |
| **Chemicals, enzymes and other reagents** | | |
| 2,3,5-Triphenyltetrazolium Chloride (TTC) | Merck | T8877 |
| 2′,7′-Dichlorodihydrofluorescein diacetate (DCFH-DA) | Merck | D6883 |
| Adenine | Merck | A3159 |
| Agar | Oxoid | L11 |

| Reagent/Resource | Reference or Source | Identifier or Catalog Number |
| --- | --- | --- |
| Antimycin A | Merck | A8674 |
| Arginine | Merck | A3609 |
| Bioanalyzer High Sensitivity DNA Assay Kit | Agilent Technologies | 5067-4626 |
| Caffeine | Merck | C0750 |
| Chelex-100 resin | Bio-Rad | 142-1253 |
| Chlorgyline | Merck | M3778 |
| Clotrimazole | Merck | C6019 |
| D-glucose | Fisher | G/0500/53 |
| DAPI | Merck | D9542 |
| Dimethylsulphoxide (DMSO) | Riedel-de Haen | 472301 |
| Fluconazole | Merck | PHR1160 |
| Formaldehyde | Sigma-Aldrich | F8775 |
| Histidine | Merck | H9511 |
| Hydrogen Peroxide | Merck | H1009 |
| Leucine | Merck | L1512 |
| LightCycler 480 SYBR Green Master Mix | Roche | 4887352001 |
| LunaScript® RT SuperMix Kit | New England Biolabs | E3010L |
| Lysine | Merck | L1211 |
| Mitotracker Green | Invitrogen | M7514 |
| Mitotracker Red CMXRos | Invitrogen | M7512 |
| Monarch® Total RNA Miniprep Kit | New England Biolabs | New England Biolabs T2010S |
| Prothioconazole-desthio | Dr Ehrenstorfer | G1287422 |
| Rhodamine 6 G | Merck | R4127 |
| Sodium Phosphate $Na_2HPO_4$ ('1 M': 89 g/L, pH'd to 7.2 with $H_3PO_4$) | Merck | 71640 |
| Tebuconazole | Dr Ehrenstorfer | C17178700 |
| Turbo DNAse (Invitrogen) | Thermo Fisher Scientific | AM2238 |
| Uracil | Merck | U1128 |
| VECTASHIELD medium | Vector Laboratories | H1000 |
| Yeast extract | DIFCO | 0886-17-0 |
| **Software** | | |
| GraphPad Prism 8 | https://www.graphpad.com/features | |
| STAR v2.7.5a | Dobin et al, 2013 https://github.com/alexdobin/STAR/releases | |
| Samtools | Li et al, 2009 https://github.com/samtools/samtools/releases/ | |
| RStudio V2023.12.1.402 | Posit team, 2024 http://www.posit.co/ | |
| Rsubread v2.16.1 (featureCounts) | Liao et al, 2014 | |
| edgeR v4.0.16 | Robinson et al, 2010 | |
| EnhancedVolcano v1.20.0 | Blighe et al, 2023 | |
| g:Profiler web tool | Raudvere et al, 2019 | |
| BioVenn web tool | Hulsen et al, 2008 | |
| ComplexHeatmaps | Gu, 2022 | |
| AutoQuantX v3.1 | Media Cybernetics https://mediacy.com/ | |
| Metamorph v7 | Molecular Devices https://www.moleculardevices.com/products/cellular-imaging-systems/high-content-analysis/metamorph-microscopy | |

| Reagent/Resource | Reference or Source | Identifier or Catalog Number |
|---|---|---|
| Nikon Ti2 Elements software | https://www.microscope.healthcare.nikon.com/products/software/nis-elements | |
| FIJI (ImageJ2 v2.14.0/1.54f) | https://imagej.net/software/fiji/ | |
| SoftMax Pro/SpecMaxMT software | Molecular Devices https://www.moleculardevices.com/products/microplate-readers/acquisition-and-analysis-software/softmax-pro-software | |
| **Other** | | |
| Illumina NextSeq platform, NextSeq 550 | Illumina | |
| Qubit fluorometer | Life-Technologies | |
| NanoDrop | Thermo Fisher Scientific | |
| Bioanalyzer system | Agilent Technologies | |
| Zeiss Axio Imager A2 fluorescence microscope | Zeiss | |
| Photometrics Prime sCMOS camera | Teledyne Photometrics | |
| Nikon Ti2 microscope | Nikon | |
| Prime 95B camera | Teledyne Photometrics | |
| Spectramax M5 Multimode plate reader | Molecular Devices https://www.moleculardevices.com/products/microplate-readers/multi-mode-readers/spectramax-m-series-readers | |

## Reagents and media

Chemicals and reagents were obtained from Merck unless indicated otherwise. Azole antifungal drugs were obtained from Ehrenstorfer. Yeast extract was from Difco. *Schizosaccharomyces pombe* cells were grown at 32 °C in Yeast Extract + Supplements (YES) medium: 0.5% w/v yeast extract, 0.2 g/l adenine, 0.2 g/l arginine, 0.2 g/l lysine, 0.2 g/l histidine, 0.2 g/l uracil, 0.2 g/l leucine. For respiratory media, the glucose in YES media was replaced with 3% glycerol, 3% glycerol + 0.1% glucose or with 2% galactose + 0.1% glucose, as described (Malecki et al, 2016). Where indicated minimal media (PMG, Pombe Minimal Glutamate media) was used (Moreno et al, 1991).

## *S. pombe* strains

Fission yeast strains employed in this study are listed in Table EV2. Standard methods for growth and manipulation of *S. pombe* were employed (Moreno et al, 1991). Deletions of genes and chromosomal regions was performed using SpEdit CRISPR-Cas9 methodology (Torres-Garcia et al, 2020a) or a selectable marker-based method (Bähler et al, 1998), using oligonucleotides listed in Table EV3. Promoter-GFP strains were constructed by first integrating the gene encoding GFP-S65T at a gene-free region on Chromosome II between the *mrp139*+ and *str1*+ genes, designated the PX2 locus, using SpEdit. Then, 0.5 kb regions from upstream of the *obr1* or *caf5* ORFs were integrated upstream of GFP. SpEdit was used to construct the *gpx1*+ overexpression strain: the *adh1* promoter and *nmt1* terminator (amplified from pRAD1 plasmid) were first integrated at the PX2 locus, followed by integration of the *gpx1*+ ORF. Constructed by SpEdit: GFP-Pap1, deletions of genes and regions within UR3, UR4, UR5, UR6 regions, *reb1Δ*, *rpm1Δ*, *hba1Δ*, *pap1Δ*, promoter-GFP reporters. Constructed by Bähler methodology (Bähler et al, 1998): *ndi1Δ*, *sdh7Δ*, *qcr7Δ*, *cox4Δ*, *atp2Δ*.

## Epimutants

The unstable/changeable nature of epimutants makes them a finite resource as expanding on selective media risks selection of cells in which further changes have occurred, including genetic changes (Torres-Garcia et al, 2020b), whereas expanding on non-selective media risks losing the resistance phenotype and epimutation. Due to limited remaining stock of UR2, other epimutants with heterochromatin islands at *cup1* were used in this study: UR20, UR28 and UR15 (Torres-Garcia et al, 2020b). For standard experiments *S. pombe* strains were thawed from glycerol stocks at −80 °C and grown on YES agar plates for 24–36 h, prior to inoculating cells in YES liquid media. Cultures were grown at 32 °C with good aeration for 12–18 h prior to performing assays.

Selection of subpopulations of epimutants: Respiratory competent (g) cells (e.g. *cup1-epi20g*): colonies were picked after growth on 2% galactose + 0.1% glucose. Respiratory-deficient (w) cells (e.g. *cup1-epi20w*) were identified by replica plating single colonies onto two YES plates, one of which was used for a TTC-overlay assay to identify white colonies (indicative of respiratory deficiency), which were picked off the unused plate. Extensive non-selective growth on YES plates (patching every 2 days for 14 days) produced *cup1-epi20yp* which had partially lost resistance to caffeine. Further non-selective growth (streaking to single colonies) produced e.g. *cup1-epi20y* which had completely lost resistance.

## Serial dilution growth assays

Equal numbers of cells five-fold were serially diluted in sterile water and spotted on various agar plates using a pinning device. Plates were incubated at 32 °C for 3 to 8 days before imaging with a flat-bed scanner at 600 dpi. Concentrations of insults and other compounds were as follows: 10, 12, 14, 16, 18, 20, 22 mM caffeine; 0.3 mM fluconazole; 0.4 µg/ml Prothioconazole-desthio (1.3 nM); 1.6 µM Tebuconazole; 50 ng/ml clotrimazole (147 nM); Antimycin A 50 ng/ml, 250 ng/ml; chlorgyline 50 µM, 150 µM. Serial dilution assays were performed at least twice and representative images shown.

## TTC assay for respiratory competence

2,3,5-Triphenyltetrazolium Chloride (TTC) is a redox indicator. Reduction of colourless TTC to red 1,3,5-triphenylformazan (TPF) is used in the tetrazolium test for metabolic activity/respiratory competence. TTC is colourless and converted to red TPF by the activities in cells competent for respiration (Nagai et al, 1961; Ogur et al, 1957; Tanaka et al, 2021). Assays were adapted for use in *S. pombe*. *Colony TTC assay*: Serially diluted cells were grown for ~3 days on YES then overlaid with 1% low melting-point agarose containing 0.5 mg/ml TTC and 100 mM sodium phosphate, pH 7.2. Plates were imaged after colour developed (typically 24 h). *Liquid TTC assay* (adapted from a method for bacteria (Defez et al, 2017)): Approximately $3.5 \times 10^8$ log-phase cells were washed twice in 10 ml phosphate buffer (50 mM sodium phosphate, pH 7.2) then pre-incubated in phosphate buffer for 30 min to deplete glucose to prevent efflux of TTC. Cells were incubated in the dark in 1 ml of 2 mg/ml TTC in 100 mM sodium phosphate for 30 min, at room temperature. After pelleting cells at $\sim 6000 \times g$, the supernatant was removed and the pellet frozen immediately on dry ice. Cells were disrupted by vortexing in 1 ml DMSO. Following centrifugation to

pellet cell debris, released TPF was measured by absorbance at 510 nm. Presented data is from three biological replicates, showing mean, standard deviation with $p$ values determined by two-tailed Student's t-test (GraphPad Prism).

## Transcriptomic analysis

Biological triplicate log phase cultures were harvested (2 ml from each) and total RNA extraction was performed using Monarch® Total RNA Miniprep Kit (New England Biolabs T2010S) according to the manufacturer's instructions, with the following modifications. Contaminating DNA was removed by 1 h incubation with Turbo DNAse (Invitrogen) followed by repeating washes and elution as per manufacturer's instructions. Quantification of RNA was performed using NanoDrop (ThermoFisher Scientific) or Qubit fluorometer (Life-Technologies) depending on downstream application. To determine the RNA transcript levels from genes of interest, quantitative reverse-transcriptase PCR (RT-qPCR) was performed. LunaScript® RT SuperMix Kit (New England Biolabs E3010L) was used for complementary DNA (cDNA) synthesis from 400 ng of RNA, following manufacturer's protocol. The kit also contains -RT control master mix that was used to generate negative controls for absence of contaminating DNA from the RNA extraction step. Quantitative analysis by qPCR in 384-well plates was carried out using a LightCycler (Roche) and LightCycler 480 SYBR Green Master Mix (Roche). Statistical tests: pairwise comparisons between wild-type and each mutant were performed (and between strains with and without Pap1), two-tailed Student's unpaired t-test in GraphPad Prism.

## RNA-sequencing (RNA-Seq)

Log phase cultures were harvested from biological triplicates of wild-type, *ppr4Δ* and *cup1-tt* cultures. Following total RNA extraction, mRNA selection was performed using NEB Next Poly(A) mRNA Magnetic Isolation Module and cDNA Library was prepared with NEB E7760 NEBNext® Ultra™ II Directional RNA Library Prep Kit for Illumina® (New England Biolabs E7490) according to manufacturer's protocol. Assessment of quality of resulting libraries was performed using an Bioanalyzer High Sensitivity DNA Assay Chip and Bioanalyzer instrument (Agilent Technologies) and quantification was performed using Qubit fluorometer (Life Technologies). After pooling, libraries were sequenced by Genome Facility Western General Hospital Edinburgh using Illumina NextSeq platform, NextSeq 550 Mid Output (130 M). Raw reads were quality assessed with FastQC and aligned and mapped to the fission yeast reference genome (972, ASM294v2, GCF_000002945.1) using STAR RNA-seq aligner and SAMtools (Dobin et al, 2013; Li et al, 2009). Generation of gene counts was performed using the featureCounts from Rsubread package and differential expression analysis was carried out using edgeR (Liao et al, 2014; Robinson et al, 2010). Volcano plots were generated using Bioconductor tool EnhancedVolcano (Blighe et al, 2023). Differential gene set analysis was prepared using g:Profiler web tool (Raudvere et al, 2019). Gene list comparisons and Venn diagrams were created using BioVenn web tool (Hulsen et al, 2008). Venn diagram $p$-values, representing probabilities that the observed overlaps occurred by chance, were determined by hypergeometric test using stats::phyper() with lower.tail =FALSE (R Core Team, 2023). Heat-maps were generated using Complex-Heatmaps (Gu, 2022).

## Chromatin immunoprecipitation (ChIP-qPCR)

ChIP was performed as previously described (Torres-Garcia et al, 2020b). For oxidative stress, GFP-Pap1 wild-type cells were treated with 0.2 mM hydrogen peroxide for 30 min prior to fixation. ChIP was performed using a rabbit anti-GFP antibody (Life Technologies, A11122). Immunoprecipitated DNA was recovered with Chelex-100 resin (Bio-Rad) and analysed by qPCR with primers targeting the promoter regions of *caf5*, *srx1*, *bfr1* and *obr1* Table EV3. ChIP-qPCR data is the mean of three biological replicates, with standard deviation reported. Statistical tests: pairwise comparisons between wild-type and each mutant were performed, two-tailed Student's unpaired t-test in GraphPad Prism.

## DCFH-DA staining for reactive oxygen species

Adapted from published methods (Jiang et al, 2021). Briefly, log-phase cells ($7 \times 10^7$) grown in YES were harvested by centrifugation, washed twice with PBS and pre-incubated in PBS for 30 min (room temperature ~25 °C), prior to incubation in 10 µM DCFH-DA for 30 min. Cells were analysed by imaging or flow cytometry. Analysis was performed at least 3 times per strain and representative images and traces are shown.

## Mitotracker staining of mitochondria

Mitotracker Red CMXRos and Mitotracker Green have, respectively, been shown to be dependent on and independent of mitochondrial membrane potential in *S. pombe* (Uehara et al, 2021). Log-phase cells grown in YES were washed twice in PMG and incubated for 45 min in PMG containing 0.5 µM Mitotracker Green and 0.25 µM Mitotracker Red CMXRos, followed by two washes in PMG and analysis by flow cytometry. At least three biological replicates were analysed and representative plots shown.

## Fluorescence microscopy

Cells were imaged on a Zeiss Axio Imager A2 fluorescence microscope equipped with a 100 Å~, 1.4-NA Plan-Apochromat objective, Chroma 86000 filter set controlled by Prior Scientific filter wheel, illumination by HBO100 mercury bulb. Image acquisition with a Photometrics Prime sCMOS camera (Teledyne Photometrics, Tucson, AZ, 85706) was controlled using Metamorph software (Version 7; Molecular Devices, San Jose, CA, 95134 USA). Pixel dimensions: 0.0623 microns/pixel. Live cells in YES medium were imaged at ~22 °C. Exposure times (FITC/ 488 nm): *obr1*^Pro^ > *GFP* cells, 200 ms; caf5pro-GFP cells, 2000 ms, or for shorter exposure 300 ms; brightfield, 1000 ms; Arg11-mCherry cells, 300 ms. For imaging of DCFH-DA staining, cells in YES media were identified using brightfield illumination and care was taken not to pre-expose cells to 488 nm light, which itself induces ROS, prior to imaging. Exposure time: 25 or 100 ms. Identical exposures were used for each sample in each set of experiments. FIJI imaging software was used: for display of sets of images, maximum intensity was determined within the set of images (FITC/GFP) and that value applied for scaling of all images. At least 3 biological replicates of each strain were imaged and representative images presented. For analysis of cells expressing GFP-Pap1, 10 ml fission yeast log phase cultures in YES were fixed for 7 min at room temperature in 3.7% formaldehyde (SIGMA, F8775), stained with DAPI and mounted in VECTASHIELD medium (Vector Labs).

Widefield fluorescence images were taken with a Nikon Ti2 microscope equipped with a Prime 95B camera (Teledyne Photometrics, Tucson, AZ 85706), using an Apo TIRF 100x/1.49 oil objective: DAPI filter, excitation 377/26, emission 432/36; GFP filter, excitation 474/26 emission 525/40 and Spectra X light-source (Lumencor, Beaverton, OR 97006 USA). For DAPI and GFP channels, a series of 13 z-slices (step size 0.3 μm) were captured to explore total cell volume. Images were deconvoluted using AutoQuantX version 3.1 (Media Cybernetics). Analysis was performed using FIJI. Briefly, images were Z-projected (max intensity) and the nuclear area was determined by thresholding DAPI channel (MaxEntropy method). The GFP signal in each cell was measured both in the nucleus and cytoplasm (nuclear region of interest (ROI) manually moved into cytoplasm). To determine Pap1 nuclear localisation this formula was employed (nuclear − cytoplasmic GFP signal)/(nuclear + cytoplasm GFP signal). At least 290 cells analysed for each genotype. Significance of the difference between samples was evaluated using Kruskal–Wallis test followed by Dunn's multiple comparison post-hoc test in GraphPad Prism.

## Flow cytometry

Flow cytometry of S. pombe cells stained with DCFH-DA was performed with a Thermo-Fisher Attune NxT flow cytometer, 488 nm laser excitation, 4 channels (channel 1 = 530/30, channel 2 = 574/26, channel 3 = 695/40, channel 4 = 780/60). Flow cytometry of S. pombe cells stained with Mitotracker dyes was performed with a Becton-Dickinson LSR Fortessa flow cytometer with Violet (405 nm), Blue (488 nm), Yellow-Green (561) and Red (640 nm) lasers; Channels B530/30 (Mitotracker Green) and Y610/20 (Mitotracker Red). Analyses and graphs were produced with FLowJo Software (FlowJo, Ashland, OR, 97520, USA).

## Rhodamine 6G efflux assay

Efflux assay was adapted from published methods for other yeast species activity (Gbelska et al, 2017). All incubations were carried out at room temperature (~25 °C). Briefly, ~2.5 × 10⁸ log-phase cells grown in YES were harvested by centrifugation, washed twice with 25 ml PBS and then incubated in 25 ml PBS for 30 min to deplete of glucose to prevent rapid efflux of Rhodamine 6G (R6G) during uptake/loading. Pelleted cells were resuspended in 4 ml PBS containing 10 μM R6G and incubated for 30 min in the dark. Excess R6G was removed with two PBS washes. Cells were provided with glucose by adding 5 ml of YES to allow efflux. 1 ml samples were taken at 2-min intervals. Samples were centrifuged at 13,000 × g in a microfuge for 45 s. 600 μl of supernatant containing effluxed R6G was removed to fresh tubes. R6G was measured with a Spectramax M5 Multimode plate reader (Molecular Devices, Wokingham), using SoftMax Pro/SpecMaxMT software, and excitation at 525 nm, auto cut-off, emission 565 nm. R6G efflux rate (arbitrary units/min) was calculated from slope, typically between 8 and 14 min. Efflux assay was performed on three biological replicates, presented mean; standard deviation with p values determined by two-tailed Student's t-test (GraphPad Prism).

## Statistical analysis and reproducibility

Sample sizes were not estimated by statistical methods; three biological replicates were used ChIP, RT-qPCR, RNA-Seq, as is standard practice in the field. No randomisation or blinding was employed. Statistical analysis was performed in Microsoft Excel and GraphPad Prism. Data were compared between two groups with unpaired two-tailed Student's t test. Data are presented as means ± SD unless indicated otherwise; P value of <0.05 considered statistically significant. For the non-normally distributed GFP-Pap1 nuclear/cytoplasm ratio data the Kruskal–Wallis test was applied, followed by Dunn's multiple comparison post-hoc test. Venn diagram p-values were determined by hypergeometric test using R. Growth assays, cytology and flow cytometry were performed at least 2 or 3 times and representative data are presented.

## Data availability

RNA-Seq data have been deposited in GEO under the accession number GSE304438. Yeast strains and other materials are available on request.

The source data of this paper are collected in the following database record: biostudies:S-SCDT-10_1038-S44318-025-00649-0.

## Peer review information

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

## Acknowledgements

We are grateful to members of the Allshire Lab, Sito Torres-Garcia and Peter Swain for valuable discussions, and to Manu Shukla and Sharon White for critical reading of the manuscript. We gratefully acknowledge David Kelly and Toni McHugh for assistance in Light Microscopy Core of the Discovery Research Platform for Hidden Cell Biology, Marie Goepp of the SBS Flow Cytometry Facility, Shaun Webb for Bioinformatics support, Martin Wear of the Edinburgh Protein Production Facility for instrumentation support, and Ken Sawin for yeast strains. We are also grateful for ongoing support from Val Wood and colleagues and the PomBase fission yeast knowledgebase. This research was funded by Wellcome via Principal Research Fellowships to RCA (200885; 224368), core funding to the Wellcome Centre for Cell Biology (203149), and the Discovery Research Platform for Hidden Cell Biology funded by Wellcome (226791). AF was funded via a School of Biological Sciences, University of Edinburgh, pilot Integrative Cell Mechanisms PhD programme studentship.

## Author contributions

**Andreas Fellas**: Conceptualization; Data curation; Formal analysis; Supervision; Validation; Investigation; Visualization; Methodology; Writing—original draft; Writing—review and editing. **Alison L Pidoux**: Conceptualization; Data curation; Formal analysis; Supervision; Funding acquisition; Validation; Investigation; Visualization; Methodology; Writing—original draft; Writing—review and editing. **Pin Tong**: Data curation; Software; Formal analysis; Validation; Visualization; Methodology. **Harriet H Hewes**: Investigation; Visualization; Methodology. **Emma C Wallace**: Investigation; Methodology. **Robin C Allshire**: Conceptualization; Supervision; Funding acquisition; Validation; Project administration; Writing—review and editing.

Source data underlying figure panels in this paper may have individual authorship assigned. Where available, figure panel/source data authorship is listed in the following database record: biostudies:S-SCDT-10_1038-S44318-025-00649-0.

## Disclosure and competing interests statement

The authors declare no competing interests.

# Expanded View Figures

**Figure EV1.   Deletion of genes within loci where heterochromatin islands are located in UR4, UR5 and UR6 epimutants.**

(**A**) Genes encompassed in heterochromatin islands present in UR4, UR5 and UR6 epimutants. Schematics of loci containing ectopic heterochromatin islands. Protein-coding genes only are indicated. Shaded blocks indicate the regions deleted in multiple-gene deletions, with expanded view below, where some genes are indicated with abbreviated names. Left: UR4 heterochromatin island, ChrII (0–60 kb); centre: UR5 heterochromatin island, ChrI (4230–4260 kb); right: UR6 heterochromatin island, ChrII (3615–3640 kb) ChIP-seq data displayed is from (Torres-Garcia et al, 2020b). (**B**) Growth assay to assess growth of deletion strains on caffeine. Single genes or regions containing multiple genes were deleted from wild-type cells and the ability of resultant strains to grow on media containing caffeine assessed. Five-fold serial dilutions of indicated strains spotted onto non-selective plates (NS; YES media) or plates containing the indicated concentrations of caffeine. Plates photographed after 2–8 days at 32 °C. (**C**) Growth assay to assess resistance to caffeine and antifungal drugs. Strains which showed some resistance to caffeine in (**B**) were retested in growth assays to assess ability to grow on various insults. Five-fold serial dilutions of indicated strains spotted onto non-selective plates (NS; YES media) or plates containing 14 mM or 16 mM caffeine (CAF), 0.3 mM fluconazole (FLC), 50 ng/ml clotrimazole (CLT). Plates photographed after 2–8 days at 32 °C.

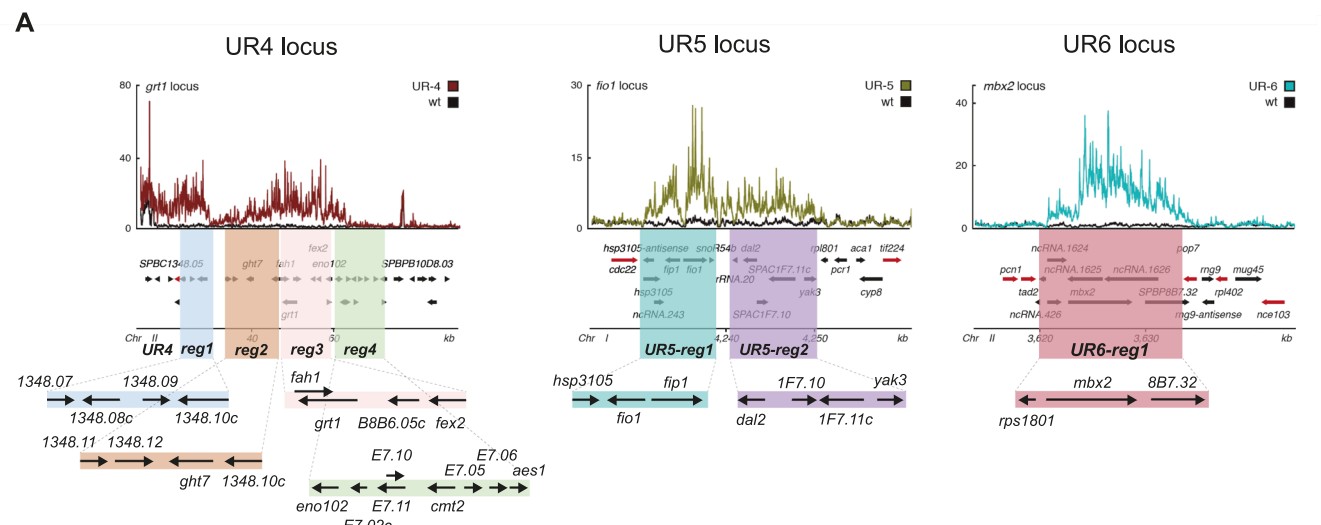

A

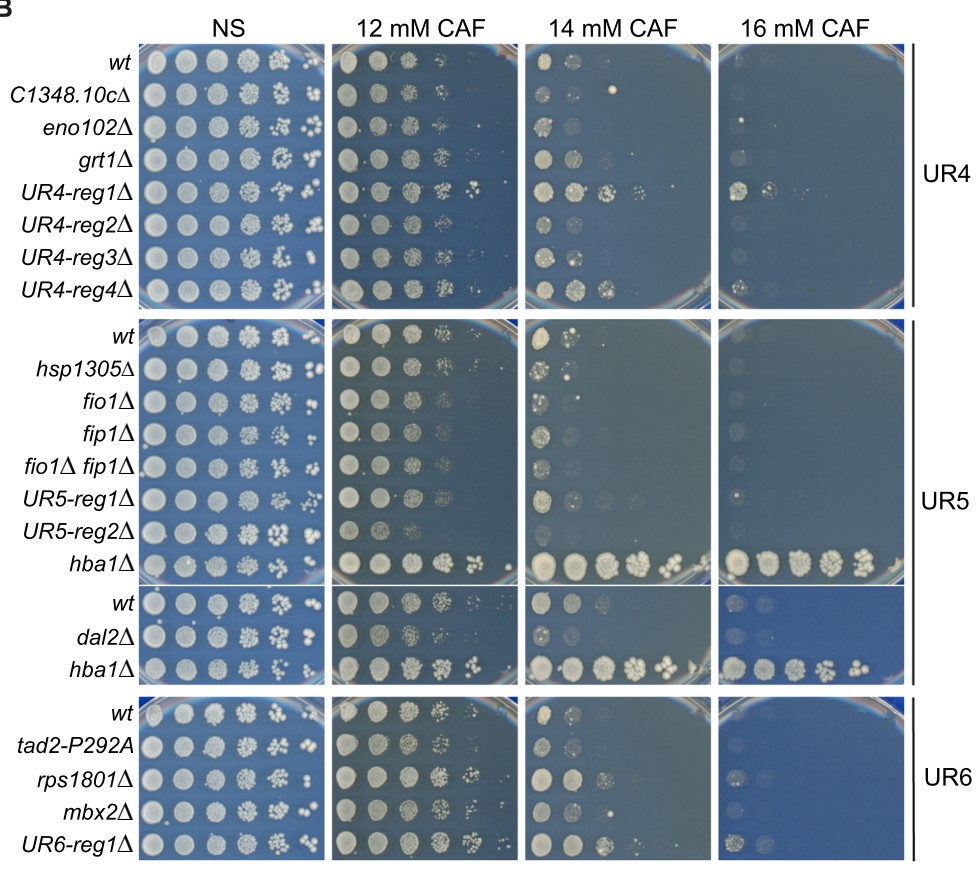

B

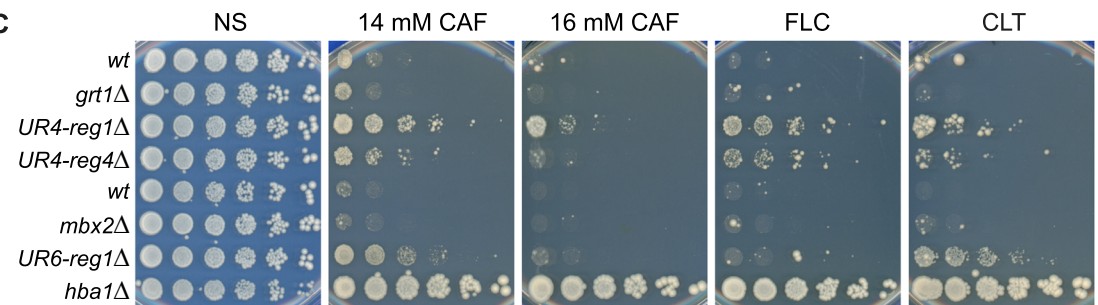

C

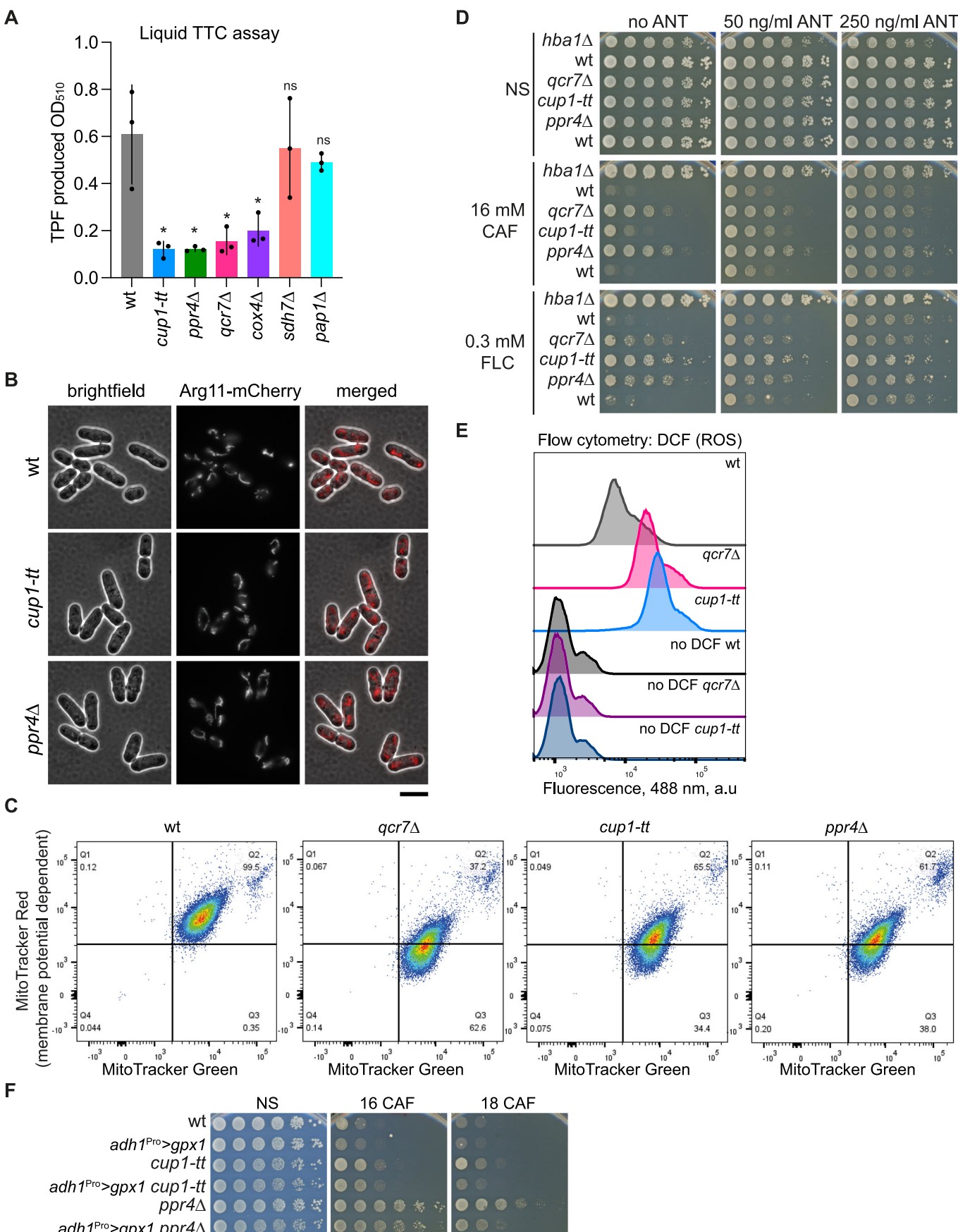

**Figure EV2. Deficiency of Cup1, Ppr4 or ETC components cause respiratory deficiency.**

(A) Liquid Tetrazolium assay for respiratory competence. Cells were incubated in 2,3,5-Triphenyltetrazolium Chloride (TTC) for 30 min before cell lysis and extraction with DMSO to release 1,3,5-triphenylformazan (TPF). TPF was measured by absorbance at 510 nm. Data are mean and standard deviation from three biological replicates. *p* values determined by two-tailed Student's t-test: *$p < 0.05$; **$p < 0.01$; ns, not significant. (B) Assessment of mitochondrial morphology. Fluorescence microscopy images of live cells of indicated strains expressing the mitochondrial protein Arg11 (Rutherford et al, 2024) tagged with mCherry (Torres-Garcia et al, 2020b). Fluorescence images are scaled relative to the brightest image. Scale bar, 10 μm. (C) Flow cytometry of cells of the indicated strains stained with Mitotracker RedCMXRos and Mitotracker Green which are respectively dependent and independent of mitochondrial membrane potential (Uehara et al, 2021). A representative experiment is shown. Percentage of cells in each quadrant are indicated: Q1, high red, low green; Q2, high red, high green; Q3, low red, high green; Q4, low red, low green. (D) Growth assay to assess impact of Antimycin A on resistance to insults. Five-fold serial dilutions of indicated strains spotted onto non-selective plates (NS; YES media containing 0.135% DMSO) or plates containing the indicated concentrations of caffeine or fluconazole, along with indicated concentrations of Antimycin A. Plates photographed after 2–8 days at 32 °C. (E) Flow cytometry of cells stained with DCHF-DA or unstained wt, *qcr7Δ* and *cup1-tt* cells are shown. (F) Growth assay to assess impact of overexpression of hydrogen peroxide scavenger Gpx1 on resistance to caffeine. Where indicated, cells contained an additional copy of *gpx1* under control of the strong *adh1* promoter. Five-fold serial dilutions of indicated strains spotted onto non-selective plates (NS; YES media) or plates containing the indicated concentrations of caffeine. Plates photographed after 2–8 days at 32 °C.

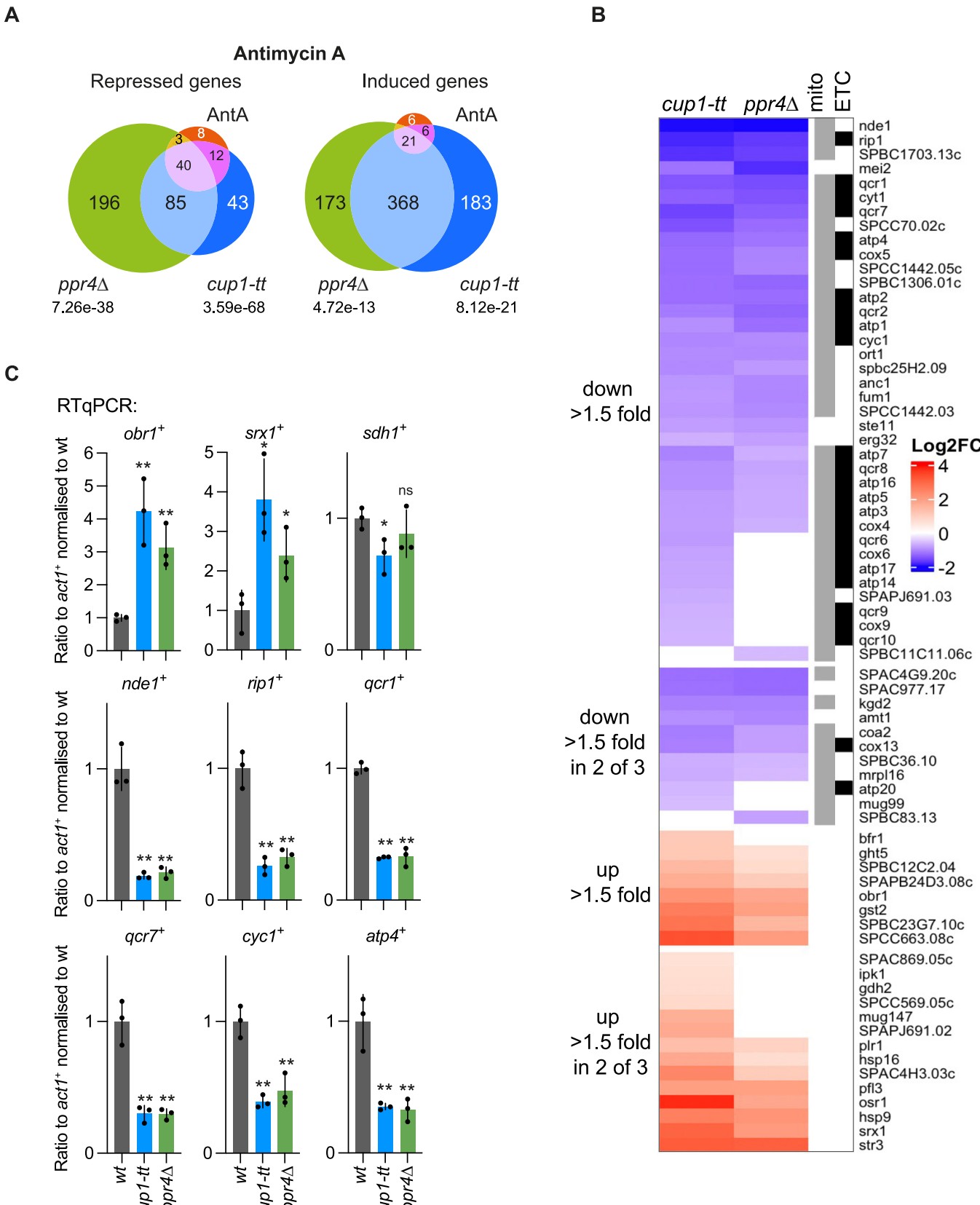

**Figure EV3.   Transcriptional profiles of cells defective for Cup1 or Ppr4 overlap with mitonuclear retrograde response.**

(A) Comparison of *cup1-tt* and *ppr4Δ* with antimycin A treatment (Malecki et al, 2016). Left: Venn diagram of comparison between genes repressed in *cup1-tt* (dark blue) and *ppr4Δ* (green) and downregulated in response to Antimycin A, which activates the mitonuclear retrograde pathway (pink). Right: Venn diagram of comparison between genes upregulated in *cup1-tt* (dark blue) and *ppr4Δ* (green) and induced in response to Antimycin A (pink). *p*-values represent probabilities that the observed overlaps occurred by chance, as determined by hypergeometric test. (B) Comparison of *cup1-tt* and *ppr4Δ* with MNR conditions. Heatmap of differentially expressed genes in *cup1-tt* vs wt and *ppr4Δ* vs wt that are altered in MNR (Malecki et al, 2016). Genes (rows) are separated into 4 sections: change (up or down) 1.5-fold in all 3 conditions (AntA treatment, *rpm1Δ* and *rep1Δ*) and changes (up or down) 1.5-fold in at least 2 out of 3 of these conditions (AntA treatment, *rpm1Δ* and *rep1Δ*), as previously described in MNR (Malecki et al, 2016). Each column represents the genotype (*cup1-tt* or *ppr4Δ* compared to wt). Only differentially expressed genes with fold-change value ≥ 1.5 and FDR-adjusted *p*-value < 0.05 from transcriptomic analysis of *cup1-tt* or *ppr4Δ* compared to wt are used in the construction of this heatmap. Each row corresponds to a gene, labelled on right. Colour gradient key represents log$_2$FC, red for upregulated and blue for downregulated transcripts (Malecki et al, 2016). Genes encoding mitochondrial proteins are indicated on the right in grey and ETC/ATP synthase subunits in black. (C) Transcript levels of ETC and antioxidant genes. Quantification by RT-qPCR of transcript levels of antioxidant genes (*obr1*$^+$, *srx1*$^+$) and nuclear-encoded ETC genes in the indicated strains. It has been shown that ETC Complex II genes such as *sdh1*$^+$ are not strongly repressed by mitochondrial dysfunction (Malecki et al, 2016). Data are mean and standard deviation from three biological replicates. *p* values determined by two-tailed Student's t-test: *$p$ < 0.05; **$p$ < 0.01; ns, not significant.

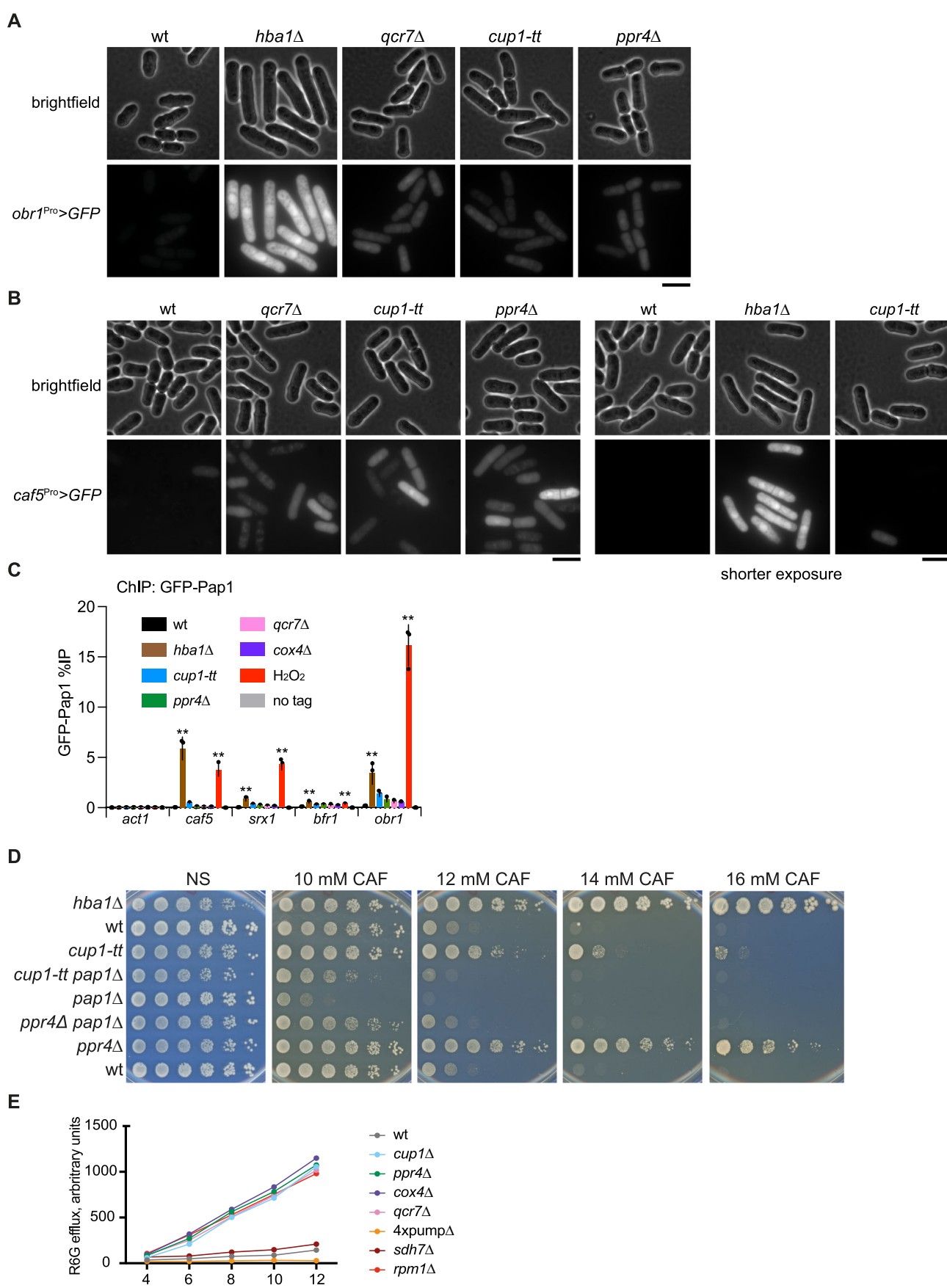

◀ **Figure EV4. Cup1, Ppr4 and ETC deficient cells activate the Pap1-dependent stress response and show increased efflux.**

(A) Pap1-dependent *obr1*-promoter reporter assay. Fluorescence microscopy images of live cells of indicated strains containing GFP under control of the Pap1-dependent *obr1* promoter. Fluorescence images are scaled relative to the brightest image (*hba1Δ*). Scale bar, 10 μm. (B) Pap1-dependent *caf5* promoter reporter assay. Fluorescence and brightfield images live cells of indicated strains containing GFP under control of the Pap1-dependent *caf1* promoter. Left panels: 2000 ms exposure. Right panels: 300 ms exposure. Fluorescence images are scaled relative to the brightest image in each set of images. Scale bar, 10 μm. (C) GFP-Pap1 ChIP-qPCR of the indicated strains and GFP-Pap1 wild-type cells treated with 0.2 mM hydrogen peroxide for 30 min. Promoter regions of the *caf5* and *bfr1* (transmembrane transporters), *srx1* (sulfiredoxin), and *obr1* (dehydrogenase) genes were analysed by qPCR to determine GFP-Pap1% immunoprecipitation. *act1* serves as negative control locus. Data are mean and standard deviation from three biological replicates. *p* values determined by two-tailed Student's t-test: *$p < 0.05$; **$p < 0.01$; ns, not significant. Only *p* values for wt vs *hba1Δ* and H$_2$O$_2$ treatment are indicated. *p* values for other mutants are shown in Fig. 5C. (D) Growth assay to assess impact of loss of Pap1 on *cup1-tt* and *ppr4Δ* mutants. Five-fold serial dilutions of indicated strains spotted onto non-selective plates (NS; YES media) or plates containing the indicated concentrations of caffeine. Plates photographed after 2–8 days at 32 °C. (E) Efflux of Rhodamine 6G (R6G) from cells. Cells of the indicated strains were preloaded with R6G and enabled to perform efflux when supplied with glucose (in YES). R6G released to the media over the indicated time-period was measured. Representative example shown. *4xpumpΔ* strain four transmembrane transporters are absent (Bfr1, Pmp1, Caf5, Mfs1) along with Pap1 and Prt1.

**A**

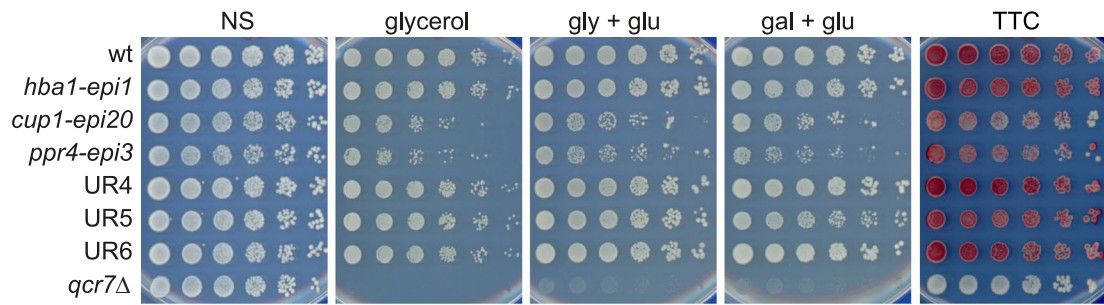

**B**

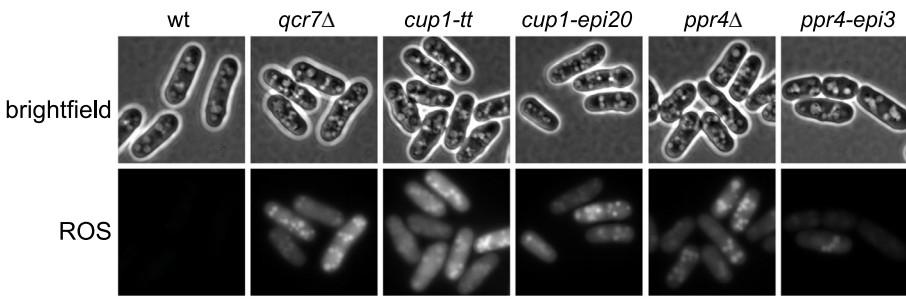

**Figure EV5.   Assessment of mitochondrial competence and ROS levels in epimutants.**

(A) Growth assay to assess respiratory competence. Five-fold serial dilutions of indicated strains spotted onto non-selective plates (NS; YES media), or YES plates in which glucose was replaced with 3% glycerol or 3% glycerol + 0.1% glucose (gly + glu) or 2% galactose + 0.1% glucose (gal + glu). Plates photographed after 2–8 days at 32 °C. TTC: after 3 days' growth colonies on YES plate were overlaid with TTC-containing agarose and incubated for ~24 h to assess respiratory competence. (B) DCHF-DA staining to assess levels of reactive oxygen species. Cells of the indicated strains were incubated in the ROS indicator DCFH-DA, which is converted to fluorescent DCF in the presence of ROS, and imaged under brightfield and 488 nm illumination. Scale bar, 10 μm.

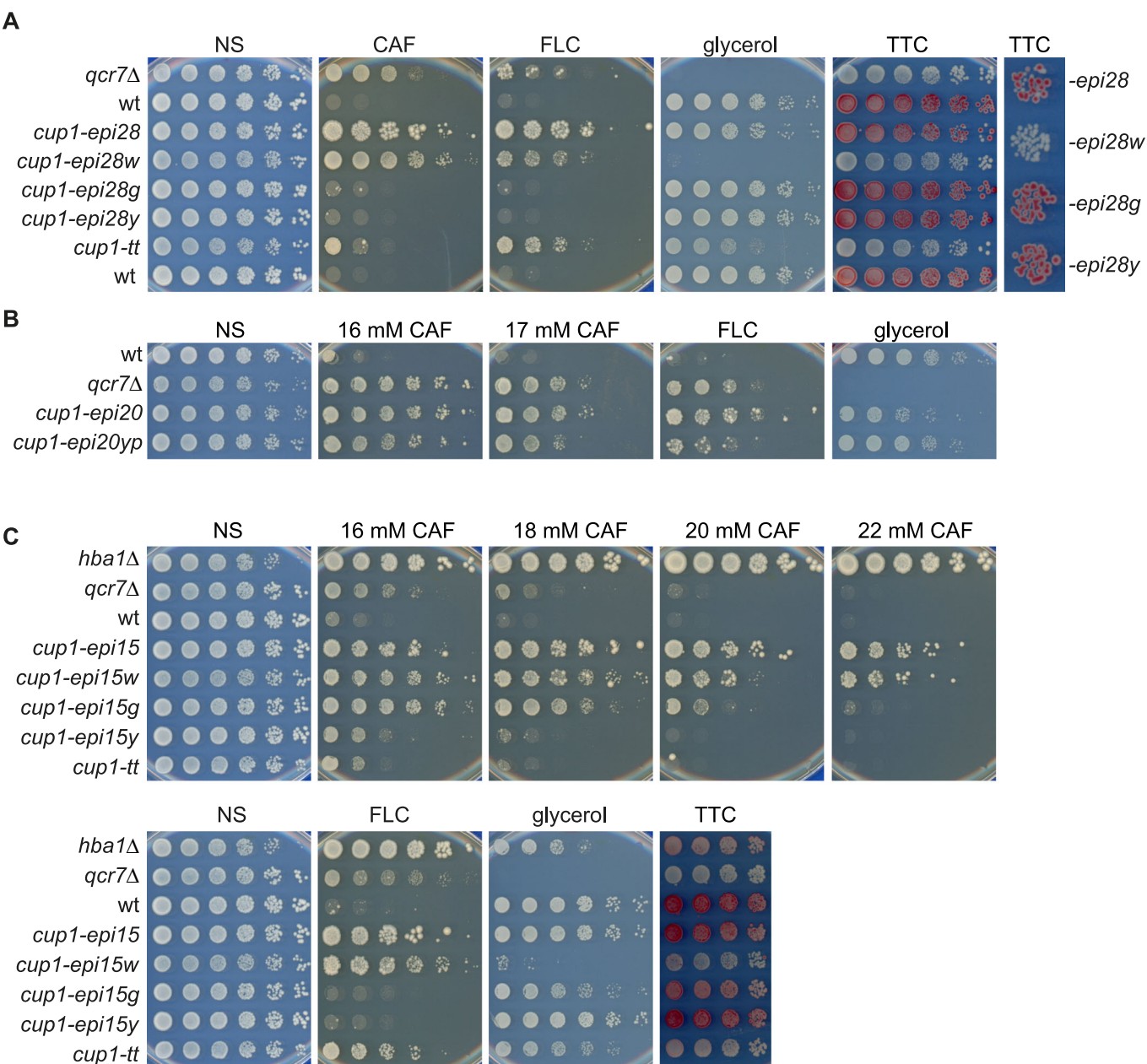

**Figure EV6. Phenotypes of *cup1* epimutant subpopulations.**

(A–C) Growth assays to assess resistance to insults and respiratory competence of isolates derived from *cup1* epimutant mixed populations: Panels show five-fold serial dilutions of indicated isolates spotted onto non-selective plates (NS; YES media) or plates containing indicated concentrations of caffeine, 0.3 mM fluconazole (FLC), YES containing 3% glycerol rather than glucose, and TTC overlay to assay respiratory competence. (A) *cup1-epi28* and derivatives; (B) *cup1-epi20* and *cup1-epi20yp* derivative; (C) *cup1-epi15* and derivatives. The NS panel is duplicated to allow comparison within each row.

