## [Peer Review File · The EMBO Journal]

Heterochromatin epimutations impose mitochondrial dysfunction to confer antifungal resistance

Andreas Fellas, Alison Pidoux, Pin Tong, Harriet Hewes, Emma Wallace, and Robin Allshire

Corresponding author(s): Robin Allshire (robin.allshire@ed.ac.uk), Alison Pidoux (Alison.Pidoux@ed.ac.uk)

Review Timeline:

Transfer from Review Commons:	24th Apr 25
Editorial Decision:	14th May 25
Appeal Received:	21st May 25
Editorial Decision:	30th Jul 25
Revision Received:	10th Sep 25
Accepted:	16th Oct 25

Editor: Cornelius Schneider

Transaction Report: This manuscript was transferred to The EMBO JOURNAL following peer review at Review Commons.

Review #1

1. Evidence, reproducibility and clarity:

Evidence, reproducibility and clarity (Required)

The manuscript under review appears to present findings on heterochromatin-mediated antifungal resistance, specifically focusing on the role of mitochondrial dysfunction in the model organism *Schizosaccharomyces pombe*. However, there are several significant concerns regarding the novelty and robustness of the conclusions drawn by the authors.

The key conclusions of the paper lack sufficient convincing evidence. While the authors attribute resistance phenotypes to heterochromatin-mediated repression, the evidence presented does not strongly support these claims. Significant claims should be qualified as preliminary or speculative, particularly those that extend beyond the experimental results provided. For example, asserting a definitive link between heterochromatin status and antifungal resistance mechanisms requires more comprehensive empirical data.

Additional experiments are crucial to bolster the claims made in the manuscript. The authors rely heavily on growth assays on nonfermentable carbon sources to supposedly elucidate respiratory function. However, this approach is outdated, and advancements in the field should be employed for a more robust assessment of mitochondrial integrity and function. Techniques such as the Seahorse assay could provide critical insights into the respiration capacity of the mutants. Furthermore, the use of electron transport chain (ETC) inhibitors like antimycin A would offer stronger evidence regarding mitochondrial dysfunction. The current use of generalized DCF staining to assess reactive oxygen species (ROS) lacks specificity. MitoSox and MitoTracker should be utilized to measure mitochondrial ROS levels and examine mitochondrial morphology effectively.

The authors claim that mitochondrial dysfunction correlates with significant changes in the transcriptome related to aerobic respiration, yet this crucial aspect lacks adequate elaboration in their analysis. Given that mitochondrial function is a primary theme of the manuscript, in-depth discussion and interpretation of the differentially expressed aerobic respiratory genes in the transcriptome data are necessary to validate their conclusions.

Additionally, as a pathogenic fungal microbiologist, I express interest in investigating whether heterochromatin-mediated resistance phenotypes are prevalent in human fungal pathogens, including *Candida albicans* and *Cryptococcus neoformans*. A bioinformatic analysis could help address this inquiry and potentially broaden the relevance of the

findings.

Lastly, in the section "Cup1 and Ppr4 deficiencies...retrograde gene repression," the conclusions are made primarily based on transcriptome analysis and lack empirical confirmation through molecular biology techniques. This section should be revised to include comprehensive molecular evidence supporting the claims.

2. Significance:

Significance (Required)

General Assessment: Strengths and Limitations

Strengths:

The study introduces a potentially novel mechanism of antifungal resistance in *Schizosaccharomyces pombe* through heterochromatin-mediated epimutations. This is particularly relevant in the context of rising antifungal resistance globally.

The focus on mitochondrial dysfunction as a contributor to drug resistance provides a deeper understanding of how fungi adapt to environmental stressors.

The manuscript raises important questions about the epigenetic factors influencing fungal resistance, which could inspire subsequent investigations in the field.

Limitations:

The research relies heavily on traditional methods for assessing respiratory function, which may not fully characterize the complexities of mitochondrial integrity and function. This may weaken the overall conclusions regarding mitochondrial dysfunction.

The evidence supporting key claims is not robust enough to confidently assert a direct link between heterochromatin changes and antifungal resistance. The lack of confirmatory experiments using more advanced techniques limits the study's impact.

The analysis of transcriptome data is insufficiently detailed, leaving significant gaps in understanding the specific mechanisms at play.

Advance: Comparison to Existing Published Knowledge

This study contributes to the existing literature by exploring the role of epimutations in antifungal resistance, aligning with emerging interests in epigenetic mechanisms in microbial adaptation. While previous studies have focused on genetic mutations and efflux mechanisms, this research attempts to link heterochromatin dynamics to resistance pathways, thereby filling a conceptual gap in understanding how eukaryotic microorganisms may adapt to antifungal treatments.

However, the advances made by the study appear to be incremental rather than groundbreaking. While it does shed light on the potential role of heterochromatin in drug resistance, further empirical evidence and a stronger methodological approach are required to substantiate these findings convincingly.

3. How much time do you estimate the authors will need to complete the suggested revisions:

Estimated time to Complete Revisions (Required)

(Decision Recommendation)

Between 3 and 6 months

4. Review Commons values the work of reviewers and encourages them to get credit for their work. Select 'Yes' below to register your reviewing activity at Web of Science Reviewer Recognition Service (formerly Publons); note that the content of your review will not be visible on Web of Science.

Yes

Review #2

1. Evidence, reproducibility and clarity:

Evidence, reproducibility and clarity (Required)

This very interesting manuscript describes the impact of heterochromatin in triggering down-regulation of mitochondrial respiratory activity in *S. pombe*, and thereby causing increased efflux and consequently increased resistance to several compounds, including

the azole class of antifungal drugs. The authors performed detailed mechanistic studies, focusing on two mitochondrial genes, *cup1* and *ppr4*, which are under heterochromatin-dependent repression. Based on their findings, they conclude that reduced mitochondrial respiration causes increased levels of reactive oxygen species (ROS), which activates the transcription factor Pap1. Pap1 then upregulates several genes, including efflux pumps. The authors performed an excellent set of experiment to address heterogenous cell populations in the epimutants, which are described in the second part of the Results section and provide strong evidence of plastic drug resistance phenotypes.

The manuscript is beautifully written and the data is presented well. Overall, the conclusions are supported by the data.

I have a reservation with one particular conclusion that I discuss below under point 1. This can be addressed by modifications to the text. Under point 2, I suggest an easy to do experiment, which would strengthen the conclusion that ROS produced due to mitochondrial dysfunction are driving the drug resistance phenotype. This is an interesting mechanism, and the data in the manuscript supports it, but the authors' have not demonstrated it directly. They could do so by using antioxidants, as I suggest below.

1. Most of the mechanistic analysis is centred around the transcription factor Pap1. The authors performed experiments to connect the production of ROS in mitochondrial mutants, with higher nuclear localisation of Pap1 and its activation of several genes, including the membrane transporter Cas5 and to a lesser extent Bfr1, which might be responsible for increased efflux. There is no question that efflux is elevated in mitochondrial mutants (a phenotype consistent with previous work in other yeast models). The authors also present data to show that inhibition of efflux reverses drug resistance. The data for Pap1's involvement is good in the *cup1* mutant (one of the mitochondrial mutants that was studied) but not so much in the *ppr4* mutant (the other mitochondrial mutant that was studied). There was little enrichment of Pap1 on the Cas5 promoter in the *ppr4* mutant, and no effects of Pap1 on the expression of Cas5 in the *ppr4* mutant (Fig 5C and 5D). While the *pap1* mutation reduced resistance of the *ppr4* mutant to drugs, the authors acknowledge that this could be due to increased sensitivity of the *pap1* mutant to drugs. The enrichment of Pap1 on the *bfr1* promoter was also modest in the mitochondrial mutants.

I would therefore suggest that another transcription factor might be responsible for the upregulation of these efflux pumps and/or other efflux pumps are involved in Pap1's contribution to drug resistance. The authors should consider modifying their conclusions

on Pap1-dependent targets that are responsible for drug resistance in the mitochondrial mutants.

2. The authors' conclusion that increased ROS levels upon dysfunctional respiration might be driving the drug resistance phenotype in *S. pombe* (via Pap1 but perhaps other mechanisms too), presents a novel mechanistic link between mitochondria and drug resistance. I would suggest solidifying this conclusion by asking if antioxidants can reduce ROS levels and thereby decrease drug resistance in *S. pombe*. N-acetyl-L-cysteine could be used for this purpose.

2. Significance:

Significance (Required)

That mitochondrial dysfunction causes drug resistance has been known for over 20 years. This manuscript describes a new mechanism, which relies on the formation of semi-stable epimutants, whereby the expression of genes encoding key mitochondrial proteins is down-regulated. As the authors propose, the beauty of epimutations is that they cause a heterogenous phenotype and are reversible, which would create an opportunity for the organism to use a bet-hedging strategy in drug. The ability to reverse the phenotype would be particularly important with using mitochondrial dysfunction as a strategy to increase drug resistance, because mitochondrial dysfunction lowers metabolic flexibility and growth rates for the organism. Therefore, it is only beneficial in the presence of drugs. This is to my knowledge one of the first logical mechanistic explanations for how fungal cells (but likely applicable more broadly) might use mitochondrial dysfunction to their advantage when needed, and then this can be reversed back to respiratory competence to maintain metabolic flexibility when drug selection is no longer present.

This study will be of high interest to researchers studying drug resistance and how phenotypic plasticity and bet-hedging mechanisms are used by cells to survive toxic compounds. This is applicable across fields. This study will further be very interesting to the fields of antifungal drug resistance and fungal pathogenesis, and will provide the foundation for studying similar mechanisms in relevant fungal pathogens of animals and plants.

My expertise is in metabolism and mitochondrial roles in fungal pathogens. I really enjoyed reading the manuscript.

3. How much time do you estimate the authors will need to complete the suggested revisions:

Estimated time to Complete Revisions (Required)

(Decision Recommendation)

Between 1 and 3 months

No

Review #3

1. Evidence, reproducibility and clarity:

Evidence, reproducibility and clarity (Required)

This study applies cellular and molecular assays, together with transcriptome analysis, to dissect how certain heterochromatin-based epimutations can confer resistance to caffeine and other drugs in fission yeast cells. The findings indicate that compromising the function of two mitochondrial proteins, Cup1 and Ppr4, leads to increased oxidants and the activation of the mito-nuclear retrograde response, which in turn causes the activation of the Pap1-mediated oxidative stress response, including the induction of transmembrane transporters to increase the efflux of drugs. This provides mechanistic insights into how the chromatin-mediated silencing of mitochondrial factors can result in fungal drug resistance. The authors also show that these phenotypes are variable within a cell population, allowing phenotypic plasticity to changing environments. This is a straightforward and clearly presented study, and the conclusions are generally justified based on the experiments presented.

****Minor comments:****

1. Fig. 3A: The legend needs more information to understand what is shown here. Does this show the normalized read counts (cpm?) for each gene scaled per average counts in all samples? Another possibility would be to show relative data for the two mutants compared

to wild-type. Also, the labels for the bottom two clusters seem the wrong way round, i.e. the last cluster should be cup1-tt only. How many genes are shown here which made the cutoff?

2. To strengthen some of the conclusions, it would be meaningful to calculate the significance of overlaps between key gene lists, given the size of the lists involved and the background gene list (Fig. 3B; Fig. 4).

3. The font size indicating the significance of differences is too tiny in some bar plots (Fig. 5C-E; Fig. 6D; Fig. 7C).

****Referees cross-commenting****

In response to issues raised by Reviewer 1:

In my opinion, the growth, TPF and ROS assays applied are robust and diagnostic to show a mitochondrial dysfunction. Additional assays, like Seahorse, would provide more specific insights about particular aspects of mitochondrial dysfunction, but this is not really relevant to this study. The key point is that the epimutations compromise mitochondrial function by downregulating mitochondrial proteins, which, in turn, are exploited by the cell to trigger a stress response that protects against antifungal compounds. The exact nature of the mitochondrial dysfunction, any changes in morphology, or details of differentially expressed genes are not critical for this mechanism, as it relies on downstream processes like the retrograde response that is activated by diverse mitochondrial problems.

The question of whether heterochromatin-mediated resistance phenotypes are prevalent in human fungal pathogens is interesting and an important avenue for future study. But it is not evident to me how this could be addressed bioinformatically.

2. Significance:

Significance (Required)

This manuscript builds on a previous study by the same group, which showed that different heterochromatin-based epimutations can provide cellular resistance to caffeine (Torres-Garcia et al. Nature, 2000). Here they use the UR1 and UR2 epimutations to highlight an example of how such mutations can generate antifungal resistance and phenotypic plasticity by exploiting side effects of mitochondrial dysfunction. Epimutations are an interesting case of cellular adaptation that lasts longer than gene-expression responses but are more readily reversible and flexible than genetic mutations, allowing bet-hedging by generating variable phenotypes in a clonal cell population. This study provides fresh

insights into the downstream effects of epimutations causing altered cellular traits, thus complementing previous studies focusing on the patterns and mechanisms of establishing heterochromatin-based genomic islands. The current study is of interest to researchers working on genome regulation, mitochondrial function, cellular adaptation/evolution, and has possible applications to combat antifungal resistance.

Field of expertise: genome regulation, gene function, fission yeast, stress response

3. How much time do you estimate the authors will need to complete the suggested revisions:

Estimated time to Complete Revisions (Required)

(Decision Recommendation)

Less than 1 month

Yes

Full Revision

Manuscript number: RC-2024-02781

Corresponding author(s): Robin Allshire & Alison Pidoux

1. General Statements [optional]

Heterochromatin epimutations impose mitochondrial dysfunction to confer antifungal resistance

Andreas Fellas, Alison L. Pidoux, Pin Tong, Harriet H. Hewes, Emma C. Wallace and Robin C. Allshire

Previously we demonstrated that fission yeast (*Schizosaccharomyces pombe*) can adapt to external insults by forming heterochromatin-dependent epimutations over regions of the genome in wild-type cells (Torres-Garcia et al, Nature 2020). These Histone H3 lysine 9 methylation-mediated heterochromatin islands reduce the expression of underlying and neighbouring genes. For two of the identified heterochromatin islands, we demonstrated that the heterochromatin-mediated repression of a specific gene in each island (*hba1⁺* and *cup1⁺*) conferred resistance to caffeine, the primary insults used, and to azole-based antifungal drugs such as fluconazole used to treat human fungal infections. Most of the resistant epimutants identified (25/30) exhibited a heterochromatin island at *cup1⁺*.

We showed that the previously uncharacterised gene *cup1⁺* encodes an essential mitochondrial LYR protein. Here we show that another of the original resistant epimutants (UR3) results from the repression of a non-essential gene encoding the mitochondrial protein Ppr4 which regulates the translation of mitochondrial genome encoded Cox1. However, it remained unknown how the reduced expression of either the Cup1 or Ppr4 mitochondrial proteins by heterochromatin islands resulted in Caffeine and Azole antifungal resistance. As epimutants are innately unstable and difficult to work with, we utilized stable genetic mutants (*cup1-tt* and *ppr4 Δ*) that mimic the resistance phenotypes *cup1-epi* and *ppr4-epi* epimutants. Our analyses are consistent with heterochromatin repression of *cup1⁺* or *ppr4⁺* imposing mitochondrial dysfunction that results in increased intracellular Reactive Oxygen Species (ROS), which, in

turn activates the canonical oxidative stress pathway. ROS causes the accumulation of oxidation-sensitive Pap1 transcription factor in the nucleus and upregulation of downstream genes encoding antioxidant activities and efflux transporters.

Remarkably, analysis of epimutants themselves indicates that their plasticity allows subpopulations to toggle between a wild-type, insult sensitive, respiration-competent, low ROS state and an insult resistant, respiration-defective, high ROS state. These distinct wild-type cellular states differ in their ability to grow on distinct carbon sources.

Resistance to antifungal agents is a huge problem humanity faces with respect to both combating human fungal pathogens that cause debilitating infections and cereal crop plant pathogens which threaten global food security. Thus, our findings have broad significance as they reveal a novel mechanism by which wild-type fungal cells can acquire unstable resistance through epimutations that cause mitochondrial dysfunction.

We look forward to hearing from you in due course.

This section is mandatory. Please insert a point-by-point reply describing the revisions that were already carried out and included in the transferred manuscript.

Our responses to each point raised by each Reviewer are presented below in blue text.

Reviewer #1 (Evidence, reproducibility and clarity (Required)):

The manuscript under review appears to present findings on heterochromatin-mediated antifungal resistance, specifically focusing on the role of mitochondrial dysfunction in the model organism *Schizosaccharomyces pombe*. However, there are several significant concerns regarding the novelty and robustness of the conclusions drawn by the authors.

The key conclusions of the paper lack sufficient convincing evidence. While the authors attribute resistance phenotypes to heterochromatin-mediated repression, the evidence presented does not strongly support these claims.

Significant claims should be qualified as preliminary or speculative, particularly those that extend beyond the experimental results provided. For example, asserting a definitive link

between heterochromatin status and antifungal resistance mechanisms requires more comprehensive empirical data.

> Our previous publication, Torres-Garcia et al, Nature, 2020 doi.org/10.1038/s41586-020-2706-x, presented extensive evidence that heterochromatin-mediated repression of genes can cause resistance to caffeine and antifungals. We reported reduced transcript levels for genes such as hba1, cup1 and ppr4 that are coated in heterochromatin in epimutants. We recapitulated heterochromatin mediation of repression to caffeine resistance by assembling synthetic heterochromatin (tetO, TetR-Clr4 system) at naïve epimutation loci. Artificial reduction of cup1 transcript levels by two additional methods also gave resistance to caffeine, as did creation of point mutation in the Cup1 LYR domain. We have expanded our description of this previous study in the Introduction to make it clear that there is compelling evidence for resistance resulting from heterochromatin-mediated repression: “Heterochromatin-mediated resistance has been recapitulated by tethering TetR-Clr4 via tetO operators to create synthetic heterochromatin at naïve epimutation loci (Torres-Garcia, 2020)”

Additional experiments are crucial to bolster the claims made in the manuscript. The authors rely heavily on growth assays on nonfermentable carbon sources to supposedly elucidate respiratory function. However, this approach is outdated, and advancements in the field should be employed for a more robust assessment of mitochondrial integrity and function. Techniques such as the Seahorse assay could provide critical insights into the respiration capacity of the mutants. Furthermore, the use of electron transport chain (ETC) inhibitors like antimycin A would offer stronger evidence regarding mitochondrial dysfunction. The current use of generalized DCF staining to assess reactive oxygen species (ROS) lacks specificity. MitoSox and MitoTracker should be utilized to measure mitochondrial ROS levels and examine mitochondrial morphology effectively.

> Growth assays are informative and used extensively in modern analyses to assess the impact of distinct environments provided by various conditions, including stresses, on manipulated yeast strains.

The purpose of this study was to test the hypothesis that mutant- or epimutant-induced mitochondrial dysfunction causes increased reactive oxygen species (ROS), leading to upregulation of membrane transporters via stress-sensing pathways such as those dependent on Pap1, leading to increased efflux and hence resistance to caffeine and azole-based antifungal drugs. In this manuscript we present evidence for each step of this mechanism. Although the basis of this mechanism is mitochondrial dysfunction, it is not necessary to acquire in-depth details of the precise mitochondrial defects, just that there is increased ROS in a) mitochondrial ETC mutants such as qcr7Δ, and b) cup1-tt and ppr4Δ. The key point of the manuscript is that the demonstrated mitochondrial dysfunction has downstream consequences that mediate resistance.

Full Revision

We therefore contend that it is unnecessary and tangential to the main focus of this study to perform Seahorse assays to determine respiratory capacity. It is sufficient to demonstrate that mutants and epimutants are respiration defective.

In-depth analysis of mitochondrial function is not main point of manuscript. The main point is that the demonstrated mito-dysfunction results in resistance to external stresses and compounds.

*Regarding Antimycin A, it is not apparent which exact experiment or hypothesis the reviewer is suggesting. The original Supplementary Figure 4 contains a Venn diagram presenting a comparison of our *cup1-tt* and *ppr4Δ* RNA-seq with published analysis (Maleki et al, 2016; DOI 10.1186/s13059-016-1101-2) of the transcriptome of cells treated with Antimycin A. The original **Figure 4A** shows a similar comparison of our RNA-seq with mutants *rpm1Δ* and *reb1Δ* which define the mitonuclear retrograde response in *S. pombe* (Maleki et al, 2016).*

*However, in response to the reviewer's interest, we also present in new Supplementary Figure **Fig. S4B** a heat-map comparison of *cup1-tt* and *ppr4Δ* with Antimycin A-treated / *rpm1Δ* / *reb1Δ* cells (i.e. those defining the mito-nuclear retrograde (MNR) pathway; Malecki 2016). This analysis is annotated to indicate genes encoding mitochondrial proteins and ETC components in particular.*

*The overlap of the transcriptional profiles between *cup1-tt*, *ppr4Δ* and Antimycin A treatment is consistent with the prediction that Antimycin A treatment could cause resistance to other insults. Indeed, we find that Antimycin A treatment allows wild-type cells to grow similarly to *cup1-tt* and *ppr4Δ* cells in the presence of caffeine or fluconazole. This data is presented in new Supplementary Figure **Fig. S2D** and supports our model that mitochondrial dysfunction (here induced by Antimycin A) leads to resistance.*

*In response the reviewer's request, we have also examined mitochondrial morphology in cells expressing the mitochondrial protein Arg11 tagged with mCherry. As shown in new Supplementary Figure **Fig. S2B**, *cup1-tt* and *ppr4Δ* cells do not show gross differences in mitochondrial morphology compared to wild-type cells, each displaying a variety of tubular and globular structures.*

*In addition, we co-stained wild-type, *cup1-tt*, *ppr4Δ* and *qcr7Δ* cells with MitoTracker Green and MitoTracker Red CMXRos. Mitotracker Red CMXRos has been used in *S. pombe* as an indicator of mitochondrial membrane potential whereas Mitotracker Green staining is not dependent on membrane potential (Uehara et al, 2021; doi.org/10.1098/rsob.200369). As shown in new Supplementary Figure **Fig. S2C**, by flow cytometry all three mutant strains have reduced red/green ratio compared to wild-type, suggesting that they have reduced membrane potential, consistent with defective mitochondrial function.*

Full Revision

Further, detailed investigations on mitochondrial morphology are tangential to the main focus and beyond the scope of this manuscript

The authors claim that mitochondrial dysfunction correlates with significant changes in the transcriptome related to aerobic respiration, yet this crucial aspect lacks adequate elaboration in their analysis. Given that mitochondrial function is a primary theme of the manuscript, in-depth discussion and interpretation of the differentially expressed aerobic respiratory genes in the transcriptome data are necessary to validate their conclusions.

*> An in-depth analysis of mitochondrial function is not the main point of manuscript. The key point is that the demonstrated mitochondrial dysfunction caused by *cup1-tt / ppr4Δ* has downstream consequences leading to resistance. We have added further analysis as described above (**Supplementary Figures S2 and S4**).*

Additionally, as a pathogenic fungal microbiologist, I express interest in investigating whether heterochromatin-mediated resistance phenotypes are prevalent in human fungal pathogens, including *Candida albicans* and *Cryptococcus neoformans*. A bioinformatic analysis could help address this inquiry and potentially broaden the relevance of the findings.

*> We refer the reviewer to this excellent review which describes chromatin readers, writers and erasers in human fungal pathogens: Buscaino, 2019; doi:10.3390/genes10110855. We have now added this reference in the Discussion, but it is beyond the scope of the current manuscript to expand our analyses to fungal pathogens. We already refer to reports of heteroresistance (unstable resistance) in *Candida* and *Cryptococcus* and speculate that epimutants could cause drug resistance in these and other fungal pathogens. In addition, we already point out in our discussion that oxidation sensitive Yap1 / Pap1 (ie AP-1-like) transcription factors are conserved in fungi, including fungal pathogens.*

We also refer this Reviewer to Reviewer 3 who argues:

****Referee 3 cross-commenting***

In response to issues raised by Reviewer 1:

The question of whether heterochromatin-mediated resistance phenotypes are prevalent in human fungal pathogens is interesting and an important avenue for future study. But it is not evident to me how this could be addressed bioinformatically."

Lastly, in the section "Cup1 and Ppr4 deficiencies...retrograde gene repression," the conclusions are made primarily based on transcriptome analysis and lack empirical confirmation through molecular biology techniques. This section should be revised to include comprehensive molecular evidence supporting the claims.

> *It is not entirely clear what the reviewer is requesting here. Our interpretation is that they are suggesting that RNA-seq alone does not qualify as molecular analysis. To confirm the RNA-seq transcriptome analysis, we have performed RT-qPCR to compare transcript levels in wt versus cup1-tt and ppr4Δ, focussing on genes known to be down-regulated by the Mito-nuclear retrograde response (MNR) conditions (as defined in S. pombe in Malecki et al 2016) along with some other relevant genes such as the dehydrogenase obr1⁺ and sulfiredoxin srx1⁺ which are upregulated by oxidative stress. As predicted, whereas most ETC-encoding transcripts are down-regulated in cup1-tt and ppr4Δ as they are in Antimycin A and rpm1Δ, reb1Δ mutants (Malecki et al 2016), Complex II sdh1⁺ transcripts are relatively unaffected. This downregulation of ETC-encoding genes, apart from Complex II is a hallmark of the MNR. The RT-qPCR analysis is presented as new Supplementary Figure Fig. S4C.*

In addition, we performed a detailed comparison of the genes down-regulated in the Malecki et al, 2016 study that defines MNR in S. pombe (i.e. Antimycin A, reb1Δ, rpm1Δ) and those up- / down-regulated in our RNA-seq of cup1-tt and ppr4Δ. This analysis is displayed as a heat-map in new Supplementary Figure Fig. S4B, and annotated to indicate nuclear genes encoding mitochondrial proteins, and the subset encoding ETC components. Again, it is apparent that there is a large overlap between the MNR genes reported by Malecki et al, 2016 and those affected in cup1-tt and ppr4Δ mutants. We conclude that both cup1-tt and ppr4Δ are indeed defective in mitochondrial function and one impact is activation of the mitonuclear retrograde pathway which has been described in other organisms, including budding yeast, to balance expression of nuclear-encoded and mitochondria-encoded mitochondrial proteins.

Reviewer #1 (Significance (Required)):

General Assessment: Strengths and Limitations

Strengths:

The study introduces a potentially novel mechanism of antifungal resistance in *Schizosaccharomyces pombe* through heterochromatin-mediated epimutations. This is particularly relevant in the context of rising antifungal resistance globally.

The focus on mitochondrial dysfunction as a contributor to drug resistance provides a deeper understanding of how fungi adapt to environmental stressors.

The manuscript raises important questions about the epigenetic factors influencing fungal resistance, which could inspire subsequent investigations in the field.

Limitations:

The research relies heavily on traditional methods for assessing respiratory function, which may not fully characterize the complexities of mitochondrial integrity and function. This may weaken the overall conclusions regarding mitochondrial dysfunction.

> It is not our aim here to fully characterise the complexities of mitochondrial integrity and function in these mutants and epimutants.

The evidence supporting key claims is not robust enough to confidently assert a direct link between heterochromatin changes and antifungal resistance. The lack of confirmatory experiments using more advanced techniques limits the study's impact.

> Our previous publication, Torres-Garcia et al, Nature, 2020 doi.org/10.1038/s41586-020-2706-x, presented extensive evidence that heterochromatin-mediated repression of genes causes resistance to caffeine and antifungals. We reported reduced transcript levels for genes such as hba1, cup1 and ppr4 that are coated in heterochromatin in epimutants. We recapitulated heterochromatin mediation of repression to caffeine resistance by assembling synthetic heterochromatin (tetO, TetR-Cir4 system) at naïve epimutation loci. Artificial reduction of cup1 transcript levels by two additional methods also gave resistance to caffeine, as did creation of point mutation in the Cup1 LYR domain. We have expanded our description of this previous study in the Introduction to make it clear that there is compelling evidence for the resistance resulting from heterochromatin-mediated repression.

An in-depth analysis of mitochondrial function is not the main point of manuscript. The key point is that the demonstrated mitochondrial dysfunction caused by cup1-tt / ppr4Δ has downstream consequences leading to resistance.

We also refer this Reviewer to Reviewer 3 who argues:

****Referees cross-commenting***

In response to issues raised by Reviewer 1:

In my opinion, the growth, TPF and ROS assays applied are robust and diagnostic to show a mitochondrial dysfunction. Additional assays, like Seahorse, would provide more specific insights about particular aspects of mitochondrial dysfunction, but this is not really relevant to this study. The key point is that the epimutations compromise mitochondrial function by downregulating mitochondrial proteins, which, in turn, are exploited by the cell to trigger a stress response that protects against antifungal compounds. The exact nature of the mitochondrial dysfunction, any changes in morphology, or details of differentially expressed genes are not critical for this mechanism, as it relies on downstream processes like the retrograde response that is activated by diverse mitochondrial problems."

Full Revision

The analysis of transcriptome data is insufficiently detailed, leaving significant gaps in understanding the specific mechanisms at play.

> *In this revised manuscript we have provided further analyses as detailed above.*

Advance: Comparison to Existing Published Knowledge

This study contributes to the existing literature by exploring the role of epimutations in antifungal resistance, aligning with emerging interests in epigenetic mechanisms in microbial adaptation. While previous studies have focused on genetic mutations and efflux mechanisms, this research attempts to link heterochromatin dynamics to resistance pathways, thereby filling a conceptual gap in understanding how eukaryotic microorganisms may adapt to antifungal treatments.

However, the advances made by the study appear to be incremental rather than groundbreaking. While it does shed light on the potential role of heterochromatin in drug resistance, further empirical evidence and a stronger methodological approach are required to substantiate these findings convincingly.

> *We thank the Reviewer for their interest in our work.*

Reviewer #2 (Evidence, reproducibility and clarity (Required)):

This very interesting manuscript describes the impact of heterochromatin in triggering down-regulation of mitochondrial respiratory activity in *S. pombe*, and thereby causing increased efflux and consequently increased resistance to several compounds, including the azole class of antifungal drugs. The authors performed detailed mechanistic studies, focusing on two mitochondrial genes, *cup1* and *ppr4*, which are under heterochromatin-dependent repression. Based on their findings, they conclude that reduced mitochondrial respiration causes increased levels of reactive oxygen species (ROS), which activates the transcription factor Pap1. Pap1 then upregulates several genes, including efflux pumps. The authors performed an excellent set of experiment to address heterogenous cell populations in the epimutants, which are described in the second part of the Results section and provide strong evidence of plastic drug resistance phenotypes.

The manuscript is beautifully written and the data is presented well. Overall, the conclusions are supported by the data.

> *We thank the Reviewer for their appreciative comments on the manuscript.*

I have a reservation with one particular conclusion that I discuss below under point 1. This can be addressed by modifications to the text. Under point 2, I suggest an easy to do experiment, which would strengthen the conclusion that ROS produced due to mitochondrial dysfunction are driving the drug resistance phenotype. This is an interesting mechanism, and the data in the manuscript supports it, but the authors' have not demonstrated it directly. They could do so by using antioxidants, as I suggest below.

1. Most of the mechanistic analysis is centred around the transcription factor Pap1. The authors performed experiments to connect the production of ROS in mitochondrial mutants, with higher nuclear localisation of Pap1 and its activation of several genes, including the membrane transporter Cas5 and to a lesser extent Bfr1, which might be responsible for increased efflux. There is no question that efflux is elevated in mitochondrial mutants (a phenotype consistent with previous work in other yeast models). The authors also present data to show that inhibition of efflux reverses drug resistance. The data for Pap1's involvement is good in the *cup1* mutant (one of the mitochondrial mutants that was studied) but not so much in the *ppr4* mutant (the other mitochondrial mutant that was studied). There was little enrichment of Pap1 on the Cas5 promoter in the *ppr4* mutant, and no effects of Pap1 on the expression of Cas5 in the *ppr4* mutant (Fig 5C and 5D). While the *pap1* mutation reduced resistance of the *ppr4* mutant to drugs, the authors acknowledge that this could be due to increased sensitivity of the *pap1* mutant to drugs. The enrichment of Pap1 on the *bfr1* promoter was also modest in the mitochondrial mutants.

I would therefore suggest that another transcription factor might be responsible for the upregulation of these efflux pumps and/or other efflux pumps are involved in Pap1's

contribution to drug resistance. The authors should consider modifying their conclusions on Pap1-dependent targets that are responsible for drug resistance in the mitochondrial mutants.

> *We agree with the Reviewer that there could be additional pathways and transcription factors involved in the upregulation of *caf5*, *bfr1* and other genes in *cup1-tt* and *ppr4Δ*. We have modified the text of the Results and Discussion to recognise the possibility that other pathways and transcriptional factors could contribute to the transcriptional response in *cup1-tt* and *ppr4Δ*.*

2. The authors' conclusion that increased ROS levels upon dysfunctional respiration might be driving the drug resistance phenotype in *S. pombe* (via Pap1 but perhaps other mechanisms too), presents a novel mechanistic link between mitochondria and drug resistance. I would suggest solidifying this conclusion by asking if antioxidants can reduce ROS levels and thereby decrease drug resistance in *S. pombe*. N-acetyl-L-cysteine could be used for this purpose.

> *We agree that this is potentially a good experiment. In fact, we had previously attempted to suppress the resistance phenotype of *cup1-tt* and *ppr4Δ* by including N-acetyl-L-cysteine or ascorbate in the media. However, we found that addition of these compounds served to increase the resistance of wild-type cells to caffeine to such an extent that we could not obtain meaningful data for *cup1-tt* and *ppr4Δ*. On the suggestion of the Reviewer, we tried other antioxidants, but exogenous glutathione did not impact resistance and Trolox had a mild effect on wild-type similar to NAC and ascorbate. We suspect that the impact of ascorbate could be due its participation in the Fenton reaction (Gradinaru & Popa, 2025; doi.org/10.3390/life15020238). In the Fenton reaction, Fe^{2+} and hydrogen peroxide react to form Fe^{3+} an hydroxyl ion and an hydroxyl radical. Ascorbate could reduce and recycle Fe^{3+} to Fe^{2+} , enabling further production of hydroxyl radicals. Hence ascorbate (and potentially other antioxidants) could act to increase rather than decrease ROS.*

*As an alternative, we sought to suppress the mutants' resistance phenotype by overexpressing an endogenous antioxidant protein in *S. pombe*. We overexpressed the hydrogen peroxide scavenger Gpx1 from the strong *adh1* promoter placed at a neutral euchromatic locus. We found that overexpression of Gpx1 had a modest impact on caffeine resistance of *cup1-tt* and *ppr4Δ*, i.e. overexpression of Gpx1 reduced the caffeine resistance phenotype of these mutants. This data is presented as new Supplementary Figure **Fig. S2F**. Thus, increasing the level of an H_2O_2 scavenger appears to reduce the resistance phenotype of *ppr4Δ* and *cup1-tt* and supports our model that ROS production due to mitochondrial dysfunction is the basis of the resistance phenotypes.*

Reviewer #2 (Significance (Required)):

That mitochondrial dysfunction causes drug resistance has been known for over 20 years. This manuscript describes a new mechanism, which relies on the formation of semi-stable

Full Revision

epimutants, whereby the expression of genes encoding key mitochondrial proteins is down-regulated. As the authors propose, the beauty of epimutations is that they cause a heterogenous phenotype and are reversible, which would create an opportunity for the organism to use a bet-hedging strategy in drug. The ability to reverse the phenotype would be particularly important with using mitochondrial dysfunction as a strategy to increase drug resistance, because mitochondrial dysfunction lowers metabolic flexibility and growth rates for the organism. Therefore, it is only beneficial in the presence of drugs. This is to my knowledge one of the first logical mechanistic explanations for how fungal cells (but likely applicable more broadly) might use mitochondrial dysfunction to their advantage when needed, and then this can be reversed back to respiratory competence to maintain metabolic flexibility when drug selection is no longer present.

This study will be of high interest to researchers studying drug resistance and how phenotypic plasticity and bet-hedging mechanisms are used by cells to survive toxic compounds. This is applicable across fields. This study will further be very interesting to the fields of antifungal drug resistance and fungal pathogenesis, and will provide the foundation for studying similar mechanisms in relevant fungal pathogens of animals and plants.

My expertise is in metabolism and mitochondrial roles in fungal pathogens. I really enjoyed reading the manuscript.

> *We thank the Reviewer for their insightful and useful comments.*

Reviewer #3 (Evidence, reproducibility and clarity (Required)):

This study applies cellular and molecular assays, together with transcriptome analysis, to dissect how certain heterochromatin-based epimutations can confer resistance to caffeine and other drugs in fission yeast cells. The findings indicate that compromising the function of two mitochondrial proteins, Cup1 and Ppr4, leads to increased oxidants and the activation of the mito-nuclear retrograde response, which in turn causes the activation of the Pap1-mediated oxidative stress response, including the induction of transmembrane transporters to increase the efflux of drugs. This provides mechanistic insights into how the chromatin-mediated silencing of mitochondrial factors can result in fungal drug resistance. The authors also show that these phenotypes are variable within a cell population, allowing phenotypic plasticity to changing environments. This is a straightforward and clearly presented study, and the conclusions are generally justified based on the experiments presented.

Minor comments:

1. Fig. 3A: The legend needs more information to understand what is shown here. Does this show the normalized read counts (cpm?) for each gene scaled per average counts in all samples? Another possibility would be to show relative data for the two mutants compared to wild-type. Also, the labels for the bottom two clusters seem the wrong way round, i.e. the last cluster should be cup1-tt only. How many genes are shown here which made the cutoff?

> *The heat-map shows differentially expressed genes with fold-change value ≥ 1.5 and FDR-adjusted p-value < 0.05 . Three biological replicates of each genotype (wt, cup1-tt and ppr4 Δ) are shown, with one replicate per column. Each row corresponds to a gene (1130 unique genes plotted). The heat-map colour gradient key represents counts per million normalised per row (z-value) i.e. each gene/sample is scaled per average counts in all 9 samples across the row.*

*We thank this Reviewer for pointing out the labelling error; the labels for the bottom two clusters were indeed the wrong way round. Upon further inspection we noticed some inconsistency in the previous plotting. We have therefore replotted the heatmap and included the dendrogram showing the hierarchical groupings of the replicates. The updated heat-map is presented in **Figure 3A** along with a more comprehensive figure legend.*

2. To strengthen some of the conclusions, it would be meaningful to calculate the significance of overlaps between key gene lists, given the size of the lists involved and the background gene list (Fig. 3B; Fig. 4).

> *We now report significance of overlaps for the data displayed in Figure 3B, Figure 4A, **Figure 4C** and Supplementary Figure **Fig S4A**. Venn diagram p-values representing probabilities that the observed pairwise overlaps occurred by chance, were determined by hypergeometric test using `stats::phyper()` with `lower.tail = FALSE`. Information has been added to the relevant figure legends and Materials & Methods.*

*In addition, in response to Reviewer 1, we performed a detailed comparison of the genes down-regulated in the Malecki et al, 2016 study that defines MNR in *S. pombe* (i.e. Antimycin A, *reb1* Δ , *rpm1* Δ) and those up- / down-regulated in our RNA-seq of *cup1-tt* and *ppr4* Δ . This analysis is displayed as a heat-map in new Supplementary Figure **Fig. S4B**, and annotated to indicate nuclear genes encoding mitochondrial proteins, and the subset encoding ETC components. Again, it is apparent that there is a large overlap between the MNR genes reported by Malecki et al, 2016 and those affected in *cup1-tt* and *ppr4* Δ mutants. We conclude that both *cup1-tt* and *ppr4* Δ are indeed defective in mitochondrial function and one impact is activation of the mitonuclear retrograde pathway.*

3. The font size indicating the significance of differences is too tiny in some bar plots (Fig. 5C-E; Fig. 6D; Fig. 7C).

> In response to the Reviewer's suggestion, we have increased the font size of the significance of differences so that the asterisks/stars are larger. To improve visibility and clarity, in cases where there is insufficient space, we have omitted not significant (ns) labelling and stated in the figure legend that unmarked columns are not significant. Where possible, we have also removed the lines/brackets and simply positioned the asterisks above the relevant column. We hope that this makes the labelling sufficiently clear.

****Referees cross-commenting****

In response to issues raised by Reviewer 1:

In my opinion, the growth, TPF and ROS assays applied are robust and diagnostic to show a mitochondrial dysfunction. Additional assays, like Seahorse, would provide more specific insights about particular aspects of mitochondrial dysfunction, but this is not really relevant to this study. The key point is that the epimutations compromise mitochondrial function by downregulating mitochondrial proteins, which, in turn, are exploited by the cell to trigger a stress response that protects against antifungal compounds. The exact nature of the mitochondrial dysfunction, any changes in morphology, or details of differentially expressed genes are not critical for this mechanism, as it relies on downstream processes like the retrograde response that is activated by diverse mitochondrial problems.

The question of whether heterochromatin-mediated resistance phenotypes are prevalent in human fungal pathogens is interesting and an important avenue for future study. But it is not evident to me how this could be addressed bioinformatically.

Full Revision

> *We agree with these comments.*

Reviewer #3 (Significance (Required)):

This manuscript builds on a previous study by the same group, which showed that different heterochromatin-based epimutations can provide cellular resistance to caffeine (Torres-Garcia et al. Nature, 2000). Here they use the UR1 and UR2 epimutations to highlight an example of how such mutations can generate antifungal resistance and phenotypic plasticity by exploiting side effects of mitochondrial dysfunction. Epimutations are an interesting case of cellular adaptation that lasts longer than gene-expression responses but are more readily reversible and flexible than genetic mutations, allowing bet-hedging by generating variable phenotypes in a clonal cell population. This study provides fresh insights into the downstream effects of epimutations causing altered cellular traits, thus complementing previous studies focusing on the patterns and mechanisms of establishing heterochromatin-based genomic islands. The current study is of interest to researchers working on genome regulation, mitochondrial function, cellular adaptation/evolution, and has possible applications to combat antifungal resistance.

> *We thank the Reviewer for their positive comments and suggested improvements.*

Field of expertise: genome regulation, gene function, fission yeast, stress response

Dear Dr. Allshire,

Thank you for submitting your manuscript "Heterochromatin epimutations impose mitochondrial dysfunction to confer antifungal resistance" which was previously reviewed at Review Commons to The EMBO Journal. I have now carefully read your study, the referee reports and your point-by-point response and discussed the manuscript with other members of the editorial team. I regret to inform you that we have decided not to pursue the publication at The EMBO Journal.

We appreciate that you investigate in detail how previously identified epimutations which cause the downregulation of mitochondrial proteins confer resistance to caffeine and azoles. The manuscript shows that the downregulation of these mitochondrial proteins leads to enhanced ROS which cause the induction of oxidative stress pathways including Pap1 which in turn leads to the expression of efflux systems that likely help to clear antifungal drugs. However, in your previous study which reported the original identification and characterization of the epimutations in fungal resistance Pap1 and its downstream pathway were already identified as an important mediator of resistance. While the data presented here further emphasize that regulation of Pap1 signaling is a central mechanism of acquired resistance we do not think that the advance is sufficiently striking to justify publication at The EMBO Journal.

Thank you in any case for giving us the opportunity to consider this manuscript. I apologize for the unusual delay in reading your manuscript which was caused by exceptional circumstances beyond my control. I hope that this will not preclude you from submitting your research to our journal in the future.

Yours sincerely,

Cornelius Schneider

Cornelius Schneider, PhD
Editor
The EMBO Journal
c.schneider@embojournal.org

Rev_Com_number: RC-2024-02781

New_manu_number: EMBOJ-2025-121174-T

Corr_author: Allshire

Title: Heterochromatin epimutations impose mitochondrial dysfunction to confer antifungal resistance

Dear Dr. Schneider,

Regarding your decision on our manuscript and the following key statement in your email below:

"However, in your previous study which reported the original identification and characterization of the epimutations in fungal resistance Pap1 and its downstream pathway were already identified as an important mediator of resistance."

We believe you have misunderstood something fundamental concerning our previous study (Torres-Garcia 2020) and perhaps this current study (Fellas, Pidoux et al). The Pap1-dependent oxidative stress triggered transcriptional response has been known to be involved in caffeine resistance for many years. In Torres-Garcia (2020) we isolated potential epimutants that displayed unstable resistance and showed that resistance results from the reduced expression of nearby genes. As a control in that study we also showed that, as expected, one stable resistant mutant isolated in the same screen had a point mutation in the gene encoding Pap1 – just to be clear, *it was not a Pap1 epimutation*.

One of our epimutants did cause heterochromatin-mediated repression of the gene encoding Hba1. The Hba1 protein was known to be required to keep Pap1 out of the nucleus in the absence of any stress. However, in most of the unstable epimutants (25/30) heterochromatin reduced the expression of a gene which we showed encoded a mitochondrial protein that we named Cup1 (Torres-Garcia 2020). It was completely unknown how resistance would result from heterochromatin-mediated repression of the cup1 gene encoding a mitochondrial protein.

In this manuscript (Fellas, Pidoux et al) we have determined the mechanism by which reduced expression of the essential *cup1* and non-essential *ppr4* genes, both encoding mitochondrial proteins, lead to unstable resistance phenotypes. Thus, we reveal how epigenetically imposed mitochondrial dysfunction can act upstream of Pap1 to provoke transcriptional changes that confer resistance.

Moreover, as we show, these epimutants exhibit variegation with respect to both resistance and respiratory competence explaining how genetically wild-type epimutant cells can exhibit both wild-type sensitivity and resistance. Such 'heteroresistance' is a key feature of fungal pathogens. We assure you, as attested to by 2 of 3 reviewers, that there has been great interest from the yeast and fungal biology communities in our finding that heterochromatin-mediated epimutants can lead to resistance by interfering with mitochondrial function and triggering the oxidative stress response.

We urge you to reconsider your decision as it appears that there may have been some misunderstanding with respect to what was and was not known or shown previously. It might be easier to discuss in a call so that we clarify any potential misunderstandings.

Yours respectfully,

Robin Allshire

and

Alison Pidoux

Dear Dr. Allshire,

Thank you for submitting a revised version of your manuscript. Your study has now been seen by two of the original referees, who find that their previous concerns have been addressed and now recommend publication of the manuscript. There remain only a few mainly editorial points that have to be addressed before I can extend formal acceptance of the manuscript:

- Please remove the figures and suppl file from the manuscript file which needs to be only text in Word and the legends of main figures should be at the end of the ms.
- On the abstract page of the manuscript, please include 4-5 general keyword terms to enhance searchability.
- Please rename the Conflict of Interest section into "Disclosure and Competing Interests Statement", in accordance with our updated Guide to Authors (<https://www.embopress.org/competing-interests>)
- As we are switching from a free-text author contribution statement towards a more formal statement based on Contributor Role Taxonomy (CRediT) terms, please remove the present Author Contribution section and instead specify each author's contribution(s) directly in the Author Information page of our submission system during upload of the final manuscript. See <https://casrai.org/credit/> for more information.
- Please adjust the format of the reference list and of the in-text citations according to EMBO Journal format (alphabetical order, author name et al + year.../up to 10 author names in the reference list before et al / please refer to our Guide to Authors for additional information on EMBO J reference format).
- Please provide either a "Yes" or a "Not Applicable" answer to each one of the questions in your Author Checklist (attached for your convenience) as you have not answered any of them yet. In the last column of this checklist, only the sections of the manuscript where the relevant information can be found should be listed (the information per se should be included in the main manuscript file).
- Please add an ORCID identifier in their respective profiles of all corresponding authors.
- Please double-check to make sure to all relevant funding information in the manuscript is also entered into our submission system. All funders acknowledged in the ms need to be in added to the system as separate entries so that the info matches in both places (the Comments box should not be used since SN can only retrieve the funding info from the separate entries)
- Please provide all main figures provided as individual production quality Figure files
- Please remove the APPENDIX FILE WITH ToC from the manuscript file, provide it as PDF and rename it to Appendix containing a Table of Contents with page numbers on the title page; the correct nomenclature should be Appendix Figure S1, etc. and Appendix Table S1, etc. throughout the file and in the ms text (callouts); there is a PDF of the suppl. figures and tables, should be the same as the one in the ms file
- Please provide suggestions for a short 'blurb' text prefacing and summing up the conceptual aspect of the study in two sentences (max. 250 characters), followed by 3-5 one-sentence 'bullet points' with brief factual statements of key results of the paper; they will form the basis of an editor-written 'Synopsis' accompanying the online version of the article. Please also provide an altered synopsis image, making sure that the aspect ratio conforms to our website's format - it should be exactly 550 pixels wide and between 300-600 pixels high.
- Please provide the Reagent and Tools Table. For more information, please check <https://www.embopress.org/page/journal/14602075/authorguide#structuredmethods> and download the template for Reagent Table
- pls Cc contact@embojournal.org so that an assistant can send the SD request to the author
- There are currently two sets of appendix files which are slightly different. Please curate these to one master set, checking carefully for figure reuse.
- Please provide a data availability statement is not in the manuscript.

*Figure Legends:

- Please note that the exact p values are not provided in the legends of figures 5B, C, D, E; 6D, 7D
- Please indicate the statistical test used for data analysis in the legend of figure 3C
- Please note that information related to n is missing in the legends of figures 3C, 5B

With best regards,

Cornelius Schneider

Cornelius Schneider, PhD
Editor | The EMBO Journal
c.schneider@embojournal.org

We realize that it is difficult to revise to a specific deadline. In the interest of protecting the conceptual advance provided by the work, we recommend a revision within 3 months (28th Oct 2025). Please discuss the revision progress ahead of this time with the editor if you require more time to complete the revisions. Use the link below to submit your revision:

Referee #1:

I read this manuscript previously for Review Commons, and provided a detailed review. The authors addressed my comments sufficiently, including with modifications to the text and new experimental data. I have no further suggestions.

Referee #2:

The authors have adequately addressed all my comments from the Review Commons assessment. I also agree with their responses to the other reviewers.

Re: Manuscript 121174

Title: Heterochromatin epimutations impose mitochondrial dysfunction to confer antifungal resistance

Authors: Andreas Fellas^{1,2,§}, Alison L. Pidoux^{1*,§}, Pin Tong¹, Harriet H. Hewes¹, Emma C. Wallace^{1,3} and Robin C. Allshire^{1*}

Referee #1:

I read this manuscript previously for Review Commons, and provided a detailed review. The authors addressed my comments sufficiently, including with modifications to the text and new experimental data. I have no further suggestions.

We thank the referee for their constructive reviewing of our manuscript.

Referee #2:

The authors have adequately addressed all my comments from the Review Commons assessment. I also agree with their responses to the other reviewers.

We thank the referee for their constructive reviewing of our manuscript.

Dear Dr. Allshire,

I am pleased to inform you that your manuscript has been accepted for publication in the EMBO Journal.

Yours sincerely,

Cornelius Schneider, PhD
Editor
The EMBO Journal
c.schneider@embojournal.org
